# An aggregon in conductin/axin2 regulates Wnt/β-catenin signaling and holds potential for cancer therapy

Dominic B. Bernkopf [1], Martina Brückner[1], Michel V. Hadjihannas [1] & Jürgen Behrens[1]

The paralogous scaffold proteins axin and conductin/axin2 are key factors in the negative regulation of the Wnt pathway transcription factor β-catenin, thereby representing interesting targets for signaling regulation. Polymerization of axin proteins is essential for their activity in suppressing Wnt/β-catenin signaling. Notably, conductin shows less polymerization and lower activity than axin. By domain swapping between axin and conductin we here identify an aggregation site in the conductin RGS domain which prevents conductin polymerization. Induction of conductin polymerization by point mutations of this aggregon results in enhanced inhibition of Wnt/β-catenin signaling. Importantly, we identify a short peptide which induces conductin polymerization via masking the aggregon, thereby enhancing β-catenin degradation, inhibiting β-catenin-dependent transcription and repressing growth of colorectal cancer cells. Our study reveals a mechanism for regulating signaling pathways via the polymerization status of scaffold proteins and suggests a strategy for targeted colorectal cancer therapy.

[1] Experimental Medicine II, Nikolaus-Fiebiger-Center, Friedrich-Alexander University Erlangen-Nürnberg, 91054 Erlangen, Germany. Correspondence and requests for materials should be addressed to D.B.B. (email: dominic.bernkopf@fau.de)

The Wnt/β-catenin signaling pathway regulates pivotal processes from pattering of body axes in early embryonic development to tissue homeostasis and stem-cell maintenance in adults[1]. Deregulation of the pathway is causally associated with severe diseases[1]. About 90% of all colorectal carcinomas, which globally represent the third leading cause for cancer-associated death[2], exhibit genetic changes activating the Wnt/β-catenin signaling pathway[3]. Under physiologic conditions, β-catenin levels are tightly regulated by the β-catenin destruction complex. The assembly of this complex is mediated by a central scaffold protein, axin, which has binding sites for β-catenin and the other complex components adenomatous polyposis coli (APC), casein kinase 1 α (CK1α), and glycogen synthase kinase 3 (GSK3)[4]. Within this complex, β-catenin is phosphorylated earmarking it for subsequent ubiquitination and proteasomal degradation[5]. The destruction complex is physiologically inhibited by Wnt ligand–receptor interactions leading to stabilization of β-catenin and its association with T-cell factor (Tcf)/lymphoid enhancer-binding factor (Lef) transcription factors in the nucleus to activate transcription of target genes[6,7]. In colorectal cancer, mutations of APC, axin or β-catenin prevent degradation of β-catenin leading to its permanent stabilization and transcription of cancer-driving target genes such as MYC and CCND1 (refs. [3,8–10]).

Conductin, also named axin2, is an axin paralog exhibiting similar domain architecture (Fig. 1a). Like axin, conductin functions as scaffold protein in the β-catenin destruction complex[11]. Moreover, AXIN2 is a β-catenin target gene[12–14], acting in a negative feedback loop to limit and fine-tune Wnt signaling[12,15]. In colorectal cancer, conductin levels are relatively high due to the constant hyperactivation of the Wnt/β-catenin signaling pathway but cannot prevent cancer growth[12].

Efficient assembly of the β-catenin destruction complex and β-catenin degradation requires homo-polymerization of axin mediated by head-to-tail polymerization of its C-terminal DIX domain[16]. This axin polymerization can be readily detected as it results in the formation of microscopically visible spherical structures, so-called axin puncta[16,17]. In contrast to axin, conductin shows no such indications for polymerization appearing diffusely distributed in the cytoplasm. In line with lower polymerization, conductin is less active in degrading β-catenin than axin[11,15]. We hypothesized that conductin-induced degradation of β-catenin might be enhanced by triggering its polymerization. By molecular comparison of axin and conductin, we show here that an aggregation site in the conductin RGS domain prevents DIX domain-mediated polymerization of conductin. Mutational inactivation of this aggregon enhances conductin polymerization and inhibition of Wnt/β-catenin signaling by conductin. Of note, conductin polymerization can also be induced by expression of a small peptide which interferes with RGS domain aggregation. Moreover, this peptide blocks β-catenin-dependent transcription and growth of colorectal cancer cells in a conductin-dependent manner suggesting potential of forced conductin polymerization for cancer therapy.

## Results

**The RGS domain prevents conductin polymerization.** As previously reported, conductin was distributed diffusely throughout the cytoplasm[11,15], whereas axin formed microscopically visible round polymers, so-called puncta in transfected cells (Fig. 1b, e)[16,17]. This difference in distribution was consistently observed in different cell lines, with different protein tags, with cDNAs from different species, and using different immunofluorescence fixation methods (Supplementary Fig. 1). By measuring staining intensities in individual cells we show that the differential distribution of axin and conductin

is independent of their respective expression levels thereby pointing to a qualitative difference between both proteins (Supplementary Fig. 2). Together our data indicate that axin polymerizes more efficiently than conductin resulting in the formation of axin puncta. To identify domains in conductin and axin responsible for differential distribution we performed domain swapping experiments. Since polymerization of axin proteins is mediated by intermolecular DIX–DIX domain interaction[16,18], we first exchanged the DIX domains between axin and conductin. Surprisingly, axin with the conductin DIX domain (Axin^CdtDIX) still formed puncta and conductin with the axin DIX domain (Cdt^AxinDIX) remained diffuse (Fig. 1b, c, e, Supplementary Fig. 3a). By generating further chimeras between axin and conductin we noticed that all proteins containing the axin RGS domain exhibited axin-like puncta, whereas all proteins containing the conductin RGS domain were distributed diffusely (Supplementary Fig. 3a, b). Importantly, the conductin RGS domain alone sufficed to prevent puncta formation of axin (Axin^CdtRGS); vice versa the axin RGS domain led to puncta formation of conductin (Cdt^AxinRGS) (Fig. 1b, c, e, Supplementary Fig. 3b). Exchange of other parts including the N-terminus, and the GSK3 and β-catenin binding domains had no impact on distribution (Supplementary Fig. 3b). These experiments demonstrate that the RGS domains alone determine the punctuate vs diffuse distribution pattern of axin and conductin, respectively. There are two principle ways how this might take place. Either the axin RGS domain promotes puncta formation or the conductin RGS domain prevents it. Further deletion analysis suggest that the latter is the case: an axin protein lacking the RGS domain (AxinΔ1–209) still formed axin-like puncta (Fig. 1b, c), whereas a conductin protein lacking the RGS domain (CdtΔRGS) lost the diffuse distribution pattern of conductin and formed puncta (Fig. 1b, c, e). Puncta formed by conductin chimera Cdt^AxinRGS and the deletion mutant CdtΔRGS represent specific DIX domain-mediated polymers because inactivation of the DIX domain by introducing a described point mutation (M3)[16] led to diffuse distribution (Fig. 1d, Supplementary Fig. 3c). Together, our findings show that the axin RGS domain is permissive for DIX-mediated polymerization, whereas the conductin RGS domain prevents polymerization.

**An aggregon within the RGS domain prevents polymerization.** Recently, cancer-associated mutations of axin were described, which induce aggregation of axin RGS domains concomitant with diffuse axin distribution and impaired β-catenin degradation[19]. This indicates that aggregation of the RGS domain prevents DIX-mediated polymerization. We speculated that conductin might contain an aggregation domain that keeps conductin permanently in a diffuse, non-polymerized state. Scanning of the conductin RGS domain for potential aggregation sites predicted by the TANGO algorithm[20] identified three sites, two of which are exclusively predicted for conductin but not axin (Fig. 2a, sites I and III) and one which is conserved between conductin and axin (Fig. 2a, site II). A possible influence on conductin polymerization was studied by individual point mutation of the aggregation sites, reasoning that abolishment of RGS aggregation should allow for DIX-mediated puncta formation. Exchange of a central tyrosine in aggregation site I to serine (Y93S), which is found at the respective position in the axin sequence, strongly reduced the TANGO score but did not lead to puncta formation (Fig. 2a–d). Likewise, a phenylalanine to arginine exchange within aggregation site II (F112R) abolished the TANGO score but did not affect diffuse distribution of conductin (Fig. 2a–d)[19]. Importantly, combined mutation of glutamine and valine to proline and serine within aggregation site III (Q188P;V189S) not only reduced the TANGO score but also activated conductin polymerization as indicated by the formation of puncta (Fig. 2a–d). These puncta

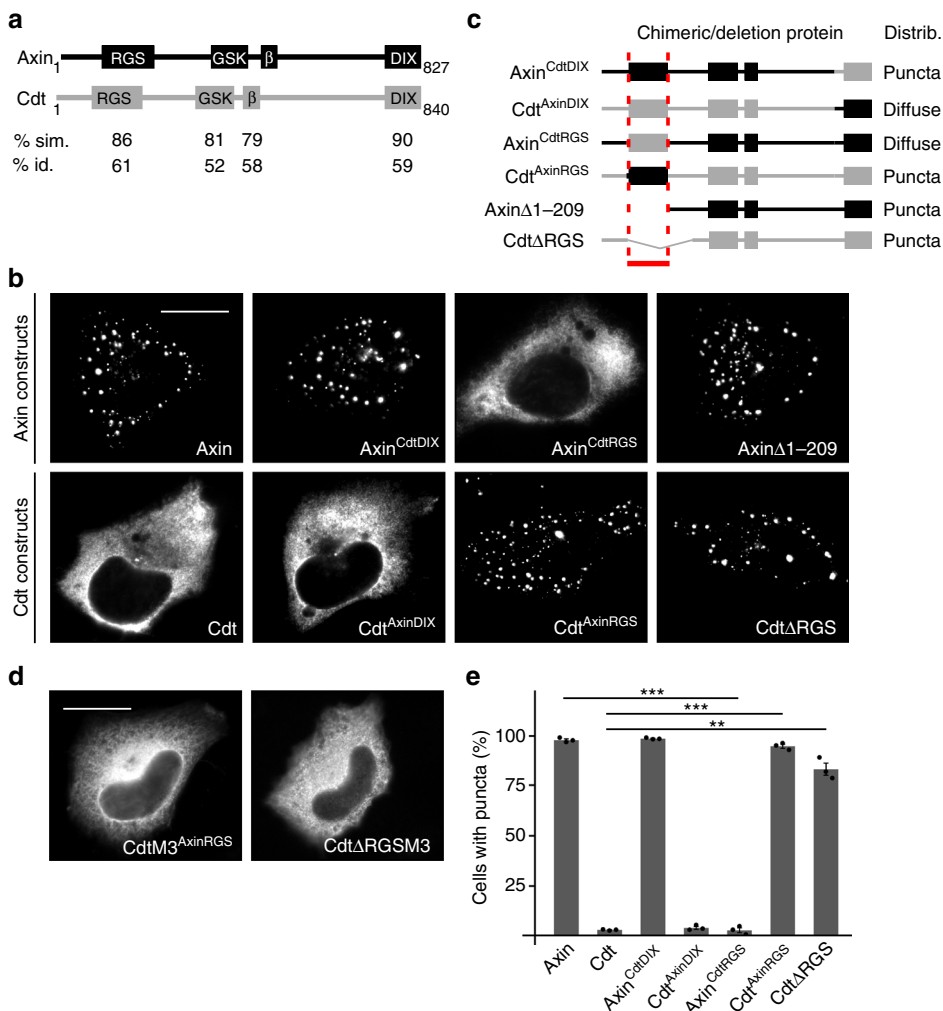

**Fig. 1** The conductin RGS domain prevents DIX-mediated polymerization. **a** Schematic to scale representation of axin and conductin (Cdt) with the domains interacting with APC (RGS), GSK3 (GSK), and β-catenin (β), and the polymerization domain (DIX). Percentage similarity (sim.) and identity (id.) are shown for each domain. **b**, **d** GFP fluorescence in U2OS cells transfected with indicated GFP-tagged axin or conductin constructs. Scale bars: 20 μm. **c** Schematic representation of chimeric axin-conductin proteins with axin parts shown in black and conductin parts in gray, and deletion mutants of axin and conductin used in **b**; not to scale. Distribution (Distrib.) is indicated on the right. Red lines mark the protein part (RGS domain) which determines distribution. **e** Percentage of transfected cells showing puncta formation of indicated constructs. Per construct, 1500 cells of three independent experiments as in **b** were analyzed. Results are mean ± SEM ($n = 3$). \*\*$p < 0.01$, \*\*\*$p < 0.001$ (Student's $t$-test). Source data are provided as a Source Data file

were dependent on DIX-mediated polymerization, since puncta formation was prevented by additional M3 mutation of the DIX domain (Fig. 2c, d). Individual mutation of glutamine to proline (Q188P) and valine to serine (V189S) was sufficient to induce polymerization (Fig. 2c), albeit with a markedly lower frequency compared to the QV to PS double mutation (Fig. 2d), indicating that both residues contribute to the activity of aggregation site III. In addition, whereas replacement of the RGS domain in axin by the conductin RGS domain (Axin^CdtRGS) abolished puncta formation, as shown above (Fig. 1b), replacement by the QV to PS mutated conductin RGS domain (Axin^CdtRGS QV-PS) did not (Fig. 2c, d). Together, the presented data suggest inhibition of DIX-mediated polymerization by aggregation site III.

Next, we reasoned that enforcing RGS aggregation of the conductin QV-PS mutant should abolish polymerization again, thereby preventing puncta formation. Since it was previously shown that mutation of leucine 106 to arginine in axin activates the otherwise silent aggregation site II[19], we introduced the corresponding L99R mutation in conductin QV-PS. Indeed, conductin L99R QV-PS was distributed diffusely again indicating

that activation of aggregation site II by L99R mutation can compensate for the inactivation of aggregation site III by QV-PS mutation (Fig. 2c, d). The already diffuse distribution of WT conductin was not affected by introducing the L99R mutation, as expected (Fig. 2c, d).

These studies were performed with mouse conductin/axin2. In line with conservation of aggregation site III, the TANGO score of human axin2 where a QM sequence replaces the QV present in conductin shows a peak at this position (Supplementary Fig. 4a). Although the TANGO score of axin2 is markedly lower compared to conductin, the change from a diffuse distribution of human axin2 to puncta formation upon QM to PS mutation demonstrated functional conservation of aggregation site III (Supplementary Fig. 4b, c). Extension of this analysis to evolutionary more distant species representing the five vertebrate classes, i.e. mammals, birds, reptiles, amphibians, and fish, revealed rather low sequence conservation of the conductin QV motif (Supplementary Fig. 5a). However, the conductin/axin2 TANGO scores of all analyzed species peak at the position of aggregation site III (Supplementary Fig. 5b). In axin, the central

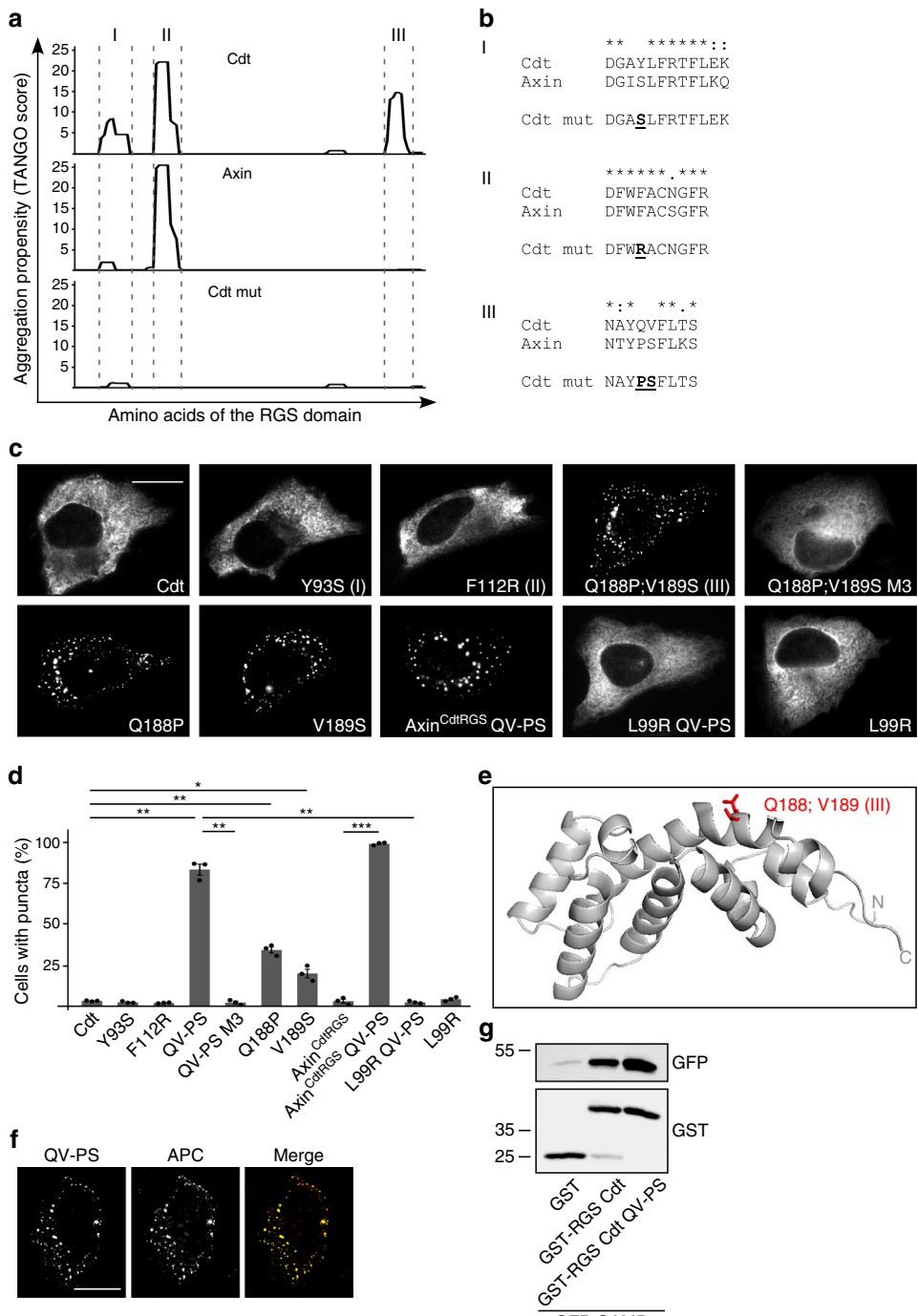

**Fig. 2** An aggregation site within the RGS domain prevents conductin polymerization. **a** Aggregation propensity score calculated by the TANGO algorithm for amino acids in the RGS domains of conductin (Cdt), axin, and mutated conductin (Cdt mut). **b** Clustal Omega alignments of conductin and axin sequences for the three aggregation sites predicted in **a**. Identity (*) and conservation between amino acid groups of strongly (:) and weakly (.) similar properties are indicated[38]. Mutated key residues are highlighted. **c** GFP fluorescence in U2OS cells transfected with indicated GFP-tagged constructs. Scale bar: 20 μm. **d** Percentage of transfected cells showing puncta formation of indicated constructs. Per construct, 1500 cells of three independent experiments as in **c** were analyzed. Results are mean ± SEM ($n = 3$). *$p < 0.05$, **$p < 0.01$, ***$p < 0.001$ (Student's $t$-test). **e** 3D structural model of the conductin RGS domain calculated by SWISS-MODEL using the crystal structure 1DK8 of the highly similar axin RGS domain (Fig. 1a) as a template[41,42]. The Global Model Quality Estimate (GMQE) reached 0.80. Amino acids mutated within aggregation site III are indicated in red. N- and C-terminus are labeled with N and C, respectively. **f** Immunofluorescence staining of APC (red) in U2OS cells transfected with GFP-tagged conductin QV-PS (green) together with APC. Scale bar: 20 μm. **g** Western blotting for indicated proteins after GST-pull down from an extract of HEK293T cells expressing GFP-SAMP which was aliquoted and supplemented with equal amounts of either GST, GST-RGS Cdt, or GST-RGS Cdt QV-PS. Source data are provided as a Source Data file

proline residue is strictly conserved among all species accompanied by an aggregation propensity of 0 at this position (Supplementary Fig. 5a, b). These data suggest that aggregation site III of conductin/axin2 is functionally conserved at least among vertebrates, and that axin aggregation is prevented, possibly by the conserved proline residue.

A 3D structural model of the conductin RGS domain based on the crystal structure of the highly similar axin RGS domain (Fig. 2e), and calculations of relative surface accessibility (Supplementary Fig. 6) showed that the key residues of aggregation site III (Q188 and V189) are exposed at the surface of the RGS domain rather than being buried within the structure. Therefore, mutation of these residues is not expected to generally destabilize the RGS domain. Furthermore, these residues were mutated into the amino acids which are present at the respective positions in axin indicating their compatibility with correct RGS domain folding (Fig. 2b). In line with this, intact folding of the QV-PS mutated RGS domain is suggested by the interaction of conductin QV-PS with the RGS-binding protein APC as seen by co-localization (Fig. 2f). In addition, pulldown of the APC RGS interaction site (SAMP) by the isolated conductin RGS domain was not affected by QV-PS mutation (Fig. 2g).

We next analyzed whether aggregation site III is functional, i.e. mediates formation of aggregates. The position of the critical QV motif of aggregation site III at the surface of the RGS domain indicates the potential to mediate protein–protein interaction (Fig. 2e, Supplementary Fig. 6). Native gel electrophoresis of a transiently expressed N-terminal conductin fragment including the RGS domain (Cdt 2–345) showed several undefined slow-migrating bands, in line with assembly of protein complexes of different sizes (Fig. 3a, upper panel). Mutational inactivation of aggregation site III (QV-PS) within this fragment resulted in faster migrating bands suggesting that aggregation site III promotes the assembly leading to slow-migrating Cdt 2–345 complexes. As a control, mutation of potential aggregation sites I or II (Y93S, F112R) did not result in faster migration of bands. Under denaturing sodium dodecyl sulfate polyacrylamide gel electrophoresis (SDS-PAGE) conditions, all fragments migrated as defined bands with similar size (Fig. 3a, lower panel). Slower migration of the WT protein compared to the QV-PS mutant in native gel electrophoresis was also observed for the isolated conductin RGS domain purified from bacteria indicating that the higher molecular weight complexes are formed by RGS–RGS aggregation independently of cellular proteins (Fig. 3b). As alternative method to show aggregation of the conductin RGS domain, sucrose density gradient ultracentrifugation was performed. Cdt 2–345 carrying the WT RGS domain sedimented efficiently penetrating into fractions of high sucrose density (Fig. 3c, fractions 16–20), which is consistent with the formation of higher molecular weight aggregates. In contrast, the Cdt 2–345 QV-PS mutant showed markedly impaired sedimentation accumulating in fractions of rather low sucrose density (Fig. 3c, fractions 8–10). These data confirm aggregation of Cdt 2–345 which is prevented by the QV-PS mutation of aggregation site III.

One application for TANGO-based aggregation prediction is to identify aggregating sequences in recombinant proteins that interfere with solubility and yield from bacterial extracts[20]. Conversely, functionality of predicted aggregation sites can be shown by lower solubility of recombinant proteins. While purifying recombinant GST-RGS Cdt, we observed poor solubility in Triton X-100-based bacterial lysates. In contrast, solubility of aggregation site mutant GST-RGS Cdt QV-PS was strongly increased (Fig. 3d, lanes 5 vs 6). GST-RGS axin showed good solubility which was abolished by the axin L106R mutation that has been previously shown to activate axin RGS aggregation (Fig. 3d, lanes 7 vs 8)[19]. Equal expression of recombinant proteins

is demonstrated by SDS-based lysis solubilizing aggregating and non-aggregating proteins (Fig. 3d, lanes 1–4).

Together our data suggest that the WT conductin RGS domain contains a functional protein–protein interaction site which causes RGS-mediated aggregation of conductin, and that this aggregon prevents DIX domain-mediated polymerization.

**Conductin polymerization augments inhibition of Wnt signaling.** Since loss of axin polymerization impairs axin-mediated β-catenin degradation[16], we determined whether gain of conductin polymerization enhances conductin-mediated inhibition of Wnt/β-catenin signaling. Degradation of β-catenin can be analyzed by expression of negative Wnt/β-catenin signaling regulators in SW480 colorectal cancer cells, which harbor high β-catenin levels due to APC mutations. As reported previously, β-catenin levels of cells expressing conductin were reduced compared to non-transfected cells (Fig. 4a)[11]. Importantly, polymerization-competent conductin QV-PS was more efficient than conductin and as efficient as axin in degrading β-catenin suggesting that the major difference between axin and conductin with respect to β-catenin degradation lies in their polymerization abilities (Fig. 4a–c). DIX domain mutation of conductin QV-PS (QV-PS M3) strongly impaired β-catenin degradation indicating the importance of DIX-mediated polymerization for β-catenin degradation by conductin QV-PS (Fig. 4a–c). Similarly, β-catenin degradation by human conductin (axin2) was enhanced by QM to PS mutation in aggregation site III (Supplementary Fig. 4d, e). In line with enhanced β-catenin degradation, conductin QV-PS was significantly more active than conductin in repressing β-catenin-dependent transcription as measured with the TOP-flash reporter assay (Fig. 4d). Again, polymerizing conductin QV-PS was as active as axin, and activity of both proteins strongly depended on DIX-mediated polymerization as indicated by loss of function of the respective DIX domain M3 mutants (Fig. 4d). Finally, we determined whether enhanced inhibition of Wnt/β-catenin signaling by conductin QV-PS translates into reduced proliferation of colorectal cancer cells. Indeed, SW480 and DLD1 cells transiently expressing conductin QV-PS proliferated significantly less compared to control cells expressing WT conductin at similar levels (Fig. 4e, Supplementary Fig. 8). In SW480 cells, inhibition of proliferation by conductin QV-PS was at least as efficient as seen for axin (Supplementary Fig. 9). Together, we could show that induction of conductin polymerization enhances β-catenin degradation and, consequently, inhibition of β-catenin-dependent transcription by conductin resulting in reduced growth of colorectal cancer cells.

**A small peptide induces polymerization of conductin.** Our data show that loss of aggregation by mutational inactivation of the aggregon induced conductin polymerization (Fig. 5a). We hypothesized that saturation of the aggregon, e.g. by binding of RGS domain-containing conductin fragments also prevents aggregation of full size conductin and allows DIX-mediated polymerization (Fig. 5a). Indeed, expression of RGS domain-containing conductin 2–345 induced polymerization of co-expressed full size conductin as seen by puncta formation (Fig. 5b, c). Partial co-localization of conductin and conductin 2–345 in puncta demonstrated interaction of both proteins. Co-expression of the aggregation dead conductin 2–345 QV-PS mutant (Fig. 3a, c) did not lead to puncta formation of full size conductin suggesting that interaction of the aggregon in the conductin fragment with full size conductin is required for puncta induction (Fig. 5b–d). Most likely, the fragment saturates the aggregon in full size conductin by interacting with it (Fig. 5a). Since the RGS domain is a rather big molecular structure

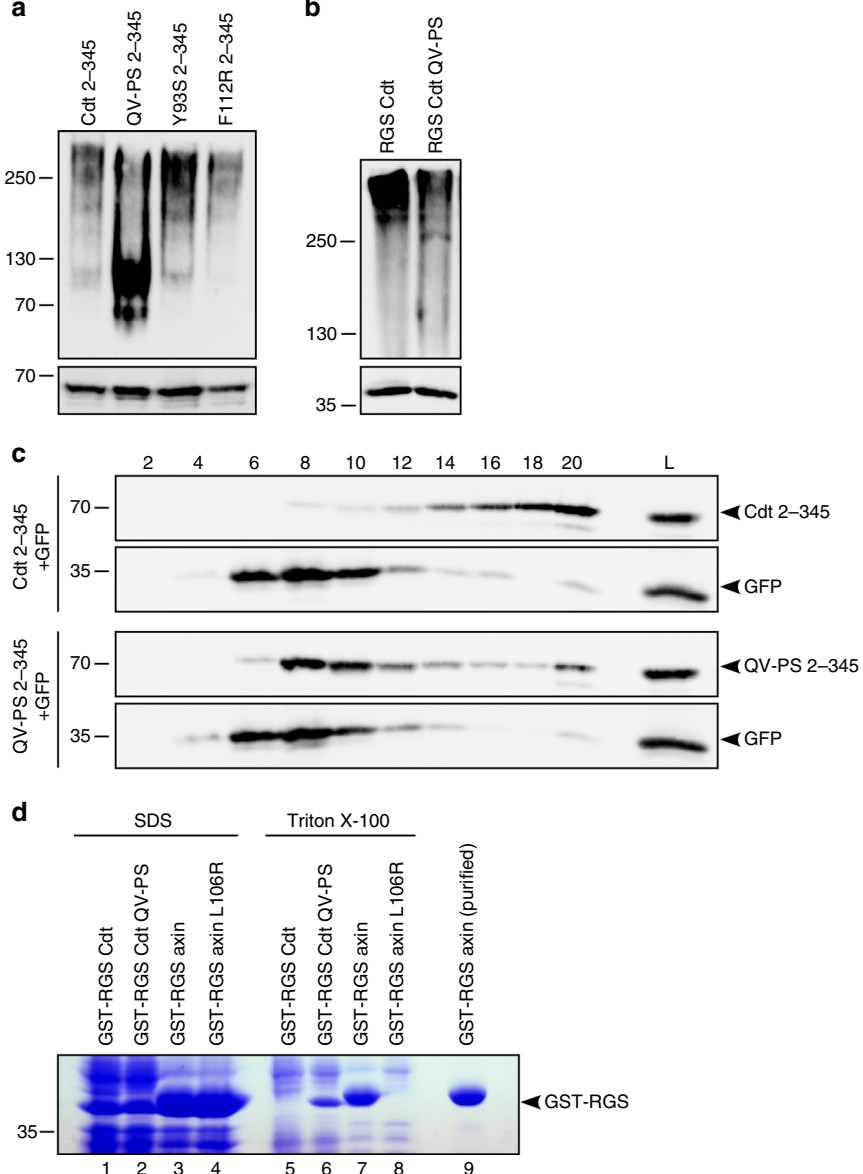

**Fig. 3** Aggregation site III is a functional interaction site (aggregon) at the RGS domain surface. **a** Western blotting for GFP under native (upper panel) or denaturing conditions (lower panel) in lysates of U2OS cells transfected with indicated GFP-tagged constructs. **b** Western blotting under native (upper panel) or denaturing conditions (lower panel) for recombinant GST-RGS Cdt and GST-RGS Cdt QV-PS purified from bacteria. **c** Western blotting for GFP in lysates (L) of HEK293T cells transfected with GFP-Cdt 2–345 (Cdt 2–345) or the GFP-Cdt 2–345 QV-PS mutant (QV-PS 2–345) together with GFP, or in fractions of these lysates prepared via ultracentrifugation through a sucrose density gradient (fractions 2 [low density] to 20 [high density]). Distribution of GFP shows comparable fractionation of both samples. **d** Coomassie Brilliant Blue staining of proteins extracted from bacteria using SDS-containing (lanes 1–4) or Triton X-100-containing (lanes 5–8) lysis buffers, or purified by pulldown on glutathione beads (lane 9). Source data are provided as a Source Data file

considering intra-cellular therapeutic applications, we tried to narrow down the sequence required for saturation to a short peptide which is centered on the aggregon and consists of conductin amino acids 182 to 195 ($P^{182-195}$: MEEN**AYQVFLTS**DI, aggregon indicated in bold). Vector-based transient expression of this 14-mer peptide $P^{182-195}$ efficiently induced polymerization of co-expressed conductin in about 80% of the cells (Fig. 5e, f). Polymerization was prevented by mutational DIX domain inactivation (conductin M3) or QV-PS mutation of the peptide suggesting that binding of the peptide aggregon to the RGS domain aggregon induces DIX-mediated polymerization of conductin (Fig. 5a, e, f). To test this hypothesis further, we studied via dot blot assays whether $P^{182-195}$ binds to the conductin RGS

domain in vitro. For this, $P^{182-195}$ was synthesized together with a C-terminal repeat of nine arginine residues ($R_9$) previously shown to confer cell permeability[21], as the peptide was designed to be applicable in cell-based experiments later on. Importantly, membrane-spotted $P^{182-195}$-$R_9$ strongly interacted with GFP-tagged proteins containing the conductin RGS domain (GFP-RGS, GFP-2–345) present in cell extracts (Fig. 5g–i). In contrast, there was neither an interaction between $P^{182-195}$-$R_9$ and GFP nor between the QV to PS mutated peptide and the RGS domain-containing proteins showing the specificity of the $P^{182-195}$-RGS interaction (Fig. 5g–i). This interaction is most likely direct because it was also observed using cell-free in vitro-translated GFP-RGS (Fig. 5j–l). Finally, $P^{182-195}$ decreased RGS-mediated

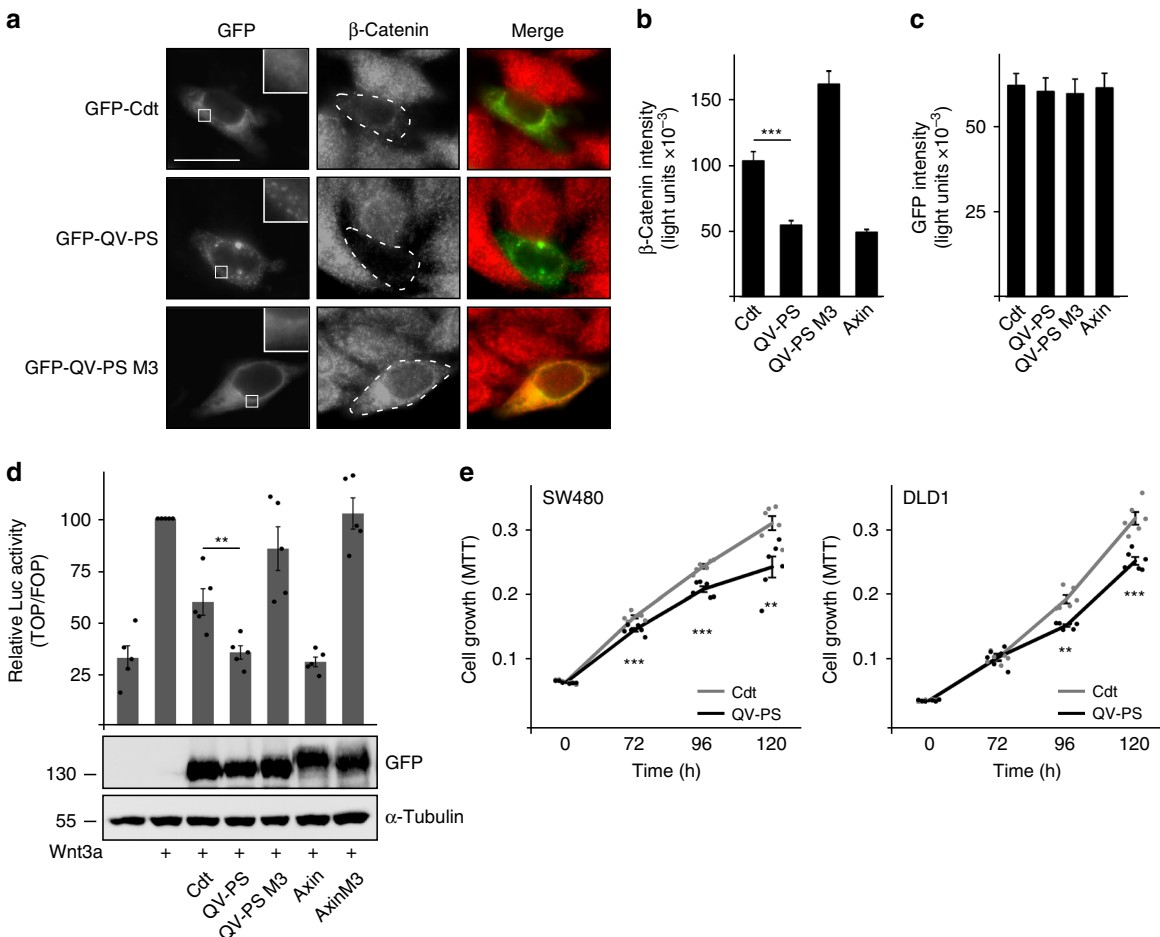

**Fig. 4** Loss of RGS aggregation increases inhibition of Wnt/β-catenin signaling by conductin. **a** Immunofluorescence staining of endogenous β-catenin (red) in SW480 cells transfected with indicated GFP-tagged constructs (green). In GFP panels, insets are shown enlarged at the upper right. Dashed lines mark transfected cells. Scale bar: 20 μm. **b** Quantification of β-catenin fluorescence intensities in one out of five representative experiments as in **a**. Results are mean ± SEM ($n = 50$). **c** Quantification of GFP intensities in cells analyzed in **b**. Results are mean ± SEM ($n = 50$). Data distribution is shown in Supplementary Fig. 7. **d** Upper panel: Luciferase activity (TOP/FOP) in HEK293T cells transfected with indicated plasmids without or with (+) Wnt3a treatment. Results are mean ± SEM ($n = 5$). Lower panel: Western blotting for GFP in extracts of HEK293T cells which were transfected with indicated GFP-tagged constructs at equal ratios as for the luciferase assay, and lysed directly in SDS-containing sample buffer due to differences in protein solubility. Loading control: α-tubulin. **e** MTT absorbance reflecting the number of viable SW480 (left panel) or DLD1 cells (right panel) expressing GFP-tagged conductin (Cdt, gray line) or the QV-PS mutant (black line) 0, 72, 96, and 120 h after seeding. One out of three representative experiments is shown. Results are mean ± SEM of six replicates ($n = 6$). **$p < 0.01$, ***$p < 0.001$ (Student's $t$-test). Source data are provided as a Source Data file

aggregation, as indicated by the reduction of higher molecular weight complexes (Fig. 5m, arrow) and the increase of lower molecular weight complexes (Fig. 5m, arrowheads) formed by the RGS domain-containing conductin fragment 2–345 upon P[182–195] co-expression. Together our experiments suggest that P[182–195] directly binds to the aggregon in the conductin RGS domain which results in inhibition of RGS-mediated aggregation and thereby promotes DIX-mediated polymerization of conductin.

**P[182–195] blocks Wnt signaling and colorectal cancer growth.** Next, we analyzed the impact of the peptide P[182–195] on Wnt/β-catenin signaling. P[182–195], but not the QV-PS mutated control peptide, significantly reduced β-catenin levels in a dosage-dependent manner after transfection in SW480 cells, which were marked by co-transfection of mScarlet-tubulin (Fig. 6a–c). Of note, P[182–195] did not reduce β-catenin levels in SW480 CRISPR/Cas9 *AXIN2* (human conductin) knockout cells showing that P[182–195] induces β-catenin degradation via axin2 (Fig. 6a–c).

Induction of β-catenin degradation was confirmed by western blotting in DLD1 and HEK293T cells, in which P[182–195] expression reduced levels of co-expressed β-catenin in a dosage-dependent manner (Fig. 6d, e). Here, β-catenin degradation was rescued by proteasome inhibition suggesting that P[182–195] enhances proteasomal degradation of β-catenin (Fig. 6d, e). Finally, when precipitating ubiquitinated proteins from cells without and with P[182–195] expression, higher levels of ubiquitinated β-catenin were detected in P[182–195]-expressing cells suggesting that P[182–195] induces ubiquitination of β-catenin (Fig. 6f, arrowheads). Thus, our data indicate that P[182–195] reduces β-catenin levels by enhancing axin2-induced ubiquitination and consequent proteasomal degradation of β-catenin.

In line with increased β-catenin degradation, expression of P[182–195] inhibited β-catenin-dependent transcription in a dosage-dependent manner, as measured by the TOP-flash reporter (Fig. 7a). Again, the QV-PS mutated peptide was much less active showing the specificity of the effect (Fig. 7b). These data suggest that the cell permeable synthetic peptide P[182–195]-R9, mentioned above, could be applicable as a therapeutic agent to inhibit Wnt

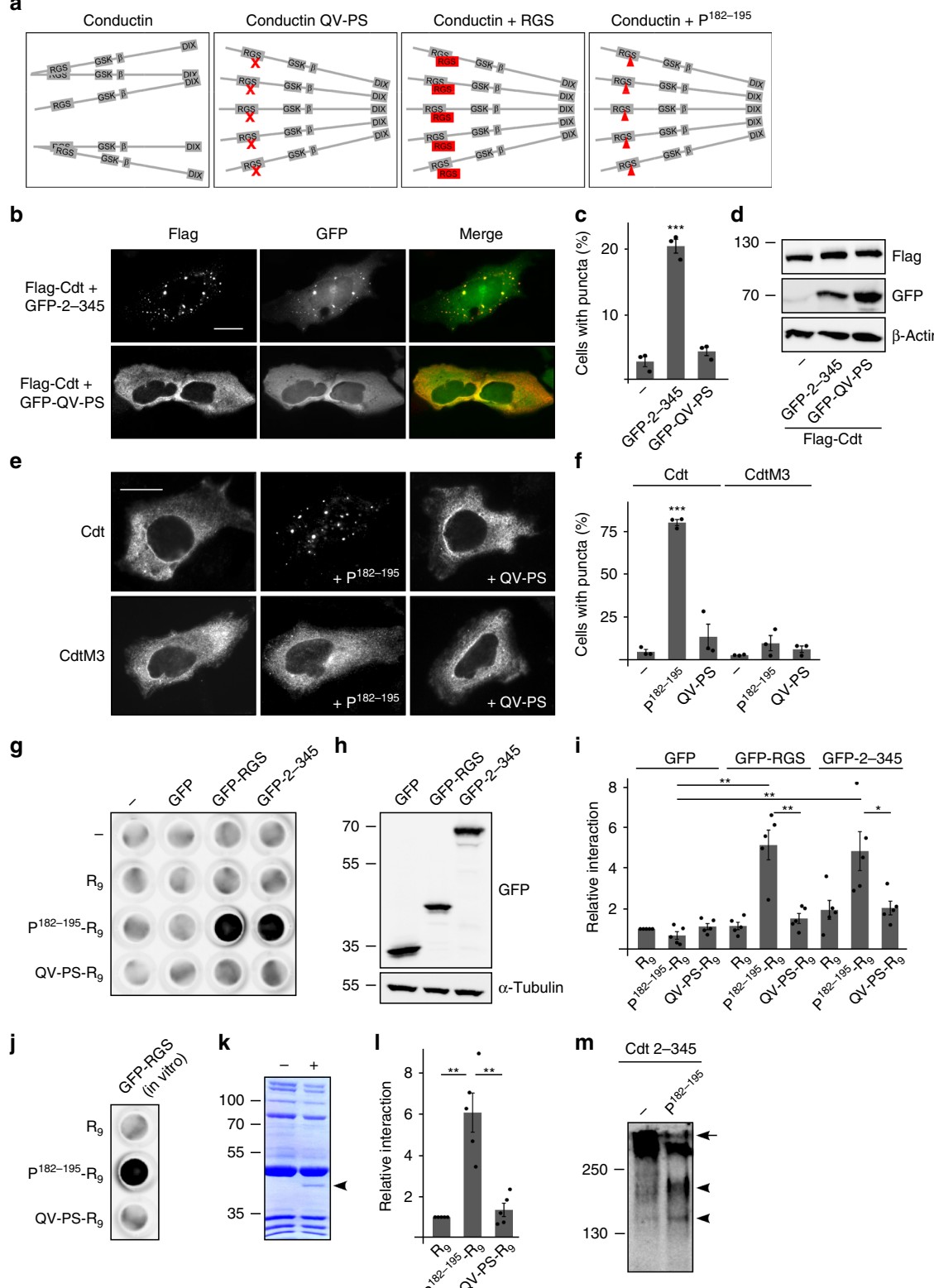

signaling in colorectal cancer cells. Indeed, in contrast to $R_9$ alone, synthetic $P^{182-195}$-$R_9$ but not the QV-PS mutated analog significantly inhibited β-catenin-dependent transcription in a dosage-dependent manner in SW480 and DLD1 cells (Fig. 7c, Supplementary Fig. 11a). SiRNA-mediated knockdown of axin2 strongly reduced inhibition of β-catenin-dependent

transcription by $P^{182-195}$ expression in SW480 and DLD1 cells (Fig. 7d, Supplementary Fig. 11b), and synthetic $P^{182-195}$-$R_9$ was almost inactive in *AXIN2* knockout cells (Fig. 7e, f) demonstrating that the peptide acts via endogenous axin2. To show that $P^{182-195}$ functions via inhibiting the axin2 aggregon, we generated SW480 CRISPR/Cas9-edited cells with QM-PS mutations in the

**Fig. 5** Saturation of the aggregon induces puncta formation of conductin. **a** Schematic illustration of conductin RGS aggregation with inhibited DIX-mediated polymerization, which can be shifted towards high-order DIX-mediated polymerization by (i) QV-PS mutation (red x), (ii) RGS co-expression, or (iii) the small peptide $P^{182-195}$ (red triangle). **b** Immunofluorescence staining for Flag (red) in U2OS cells transfected with Flag-tagged conductin (Flag-Cdt) either together with conductin 2–345 or conductin 2–345 QV-PS tagged with GFP (green). Scale bar: 20 μm. **c** Percentage of transfected cells exhibiting Cdt puncta. Per bar, 1500 cells of three independent experiments as in **b** were analyzed. Results are mean ± SEM ($n = 3$). **d** Western blotting for indicated proteins in U2OS cell extracts. **e** Immunofluorescence staining of conductin or its M3 mutant in U2OS cells transfected with HA-tagged conductin or conductin M3 either alone, or together with $P^{182-195}$, or the QV-PS mutant of the peptide. Scale bar: 20 μm. **f** Percentage of transfected cells exhibiting Cdt or CdtM3 puncta. Per bar, 900 cells of three independent experiments as in **e** were analyzed. Results are mean ± SEM ($n = 3$). **g, j** Dot blot: Detection of GFP(-tagged) proteins binding to membrane pieces which were spotted with $H_2O$ (−) or 8 nmol of $R_9$, $P^{182-195}$-$R_9$ or the QV-PS mutated peptide (QV-PS-$R_9$) prior to incubation with lysates of HEK293T cells transfected with indicated plasmids (**g**) or with in vitro translated GFP-RGS (**j**). **h** Western blotting for GFP in the lysates used in **g**. Loading control: α-tubulin. **i, l** 2D densitometry quantification of dot blots in **g** and **j**, respectively. Results are presented relative to the GFP/$R_9$ (**i**) or the GFP-RGS/$R_9$ (**l**) combination as mean ± SEM of five independent dot blots ($n = 5$). **k** Coomassie Brilliant Blue staining of in vitro translated GFP-RGS (+DNA template lane, arrowhead) used in **j**. Bands also present without template DNA (−) show purified kit components. **m** Western blotting for HA under native conditions in lysates of HEK293T cells transfected with HA-Cdt 2–345 alone (−) or together with $P^{182-195}$. *$p < 0.05$, **$p < 0.01$, ***$p < 0.001$ (Student's $t$-test). Source data are provided as a Source Data file

aggregon of axin2. In these clones, the peptide was significantly less active in repressing the TOP reporter than in WT control cells indicating that the peptide activates axin2 via interacting with the aggregon (Fig. 7g).

Consistent with inhibition of β-catenin-dependent transcription in reporter assays, synthetic $P^{182-195}$-$R_9$ also inhibited expression of β-catenin target genes in DLD1 and SW480 cells but not in SW480 *AXIN2* knockout cells (Fig. 7h, Supplementary Fig. 11c). Importantly, expression of $P^{182-195}$ reduced growth of the colorectal cancer cells SW480 in a colony formation assay in a dosage-dependent manner (Fig. 7i, j). Moreover, treatment with synthetic $P^{182-195}$-$R_9$ significantly reduced colony formation of DLD1 and SW480 cells but not of SW480 *AXIN2*-knockout cells compared to $R_9$-treated control cells, most likely through inhibiting Wnt/β-catenin signaling via axin2 (Fig. 7k, l, Supplementary Fig. 11d, e). The QV-PS mutated control peptide, which cannot induce conductin polymerization (Fig. 5f), was far less active in inhibiting colony formation (Fig. 7i–l, Supplementary Fig. 11d, e).

Since tankyrase inhibitors block Wnt/β-catenin signaling by increasing the amounts of axin2/conductin and axin[22], we analyzed whether the peptide and the tankyrase inhibitor G007-LK cooperate to inhibit Wnt/β-catenin signaling. Indeed, while $P^{182-195}$ expression from 5 ng vector or treatment with 50 nM G007-LK individually inhibited β-catenin-dependent transcription by about 40%, the combination of both inhibited by about 75%, similar to the inhibition by 100 ng vector alone (Fig. 7a, m).

## Discussion

Axin and conductin act in a similar way as scaffolds for downregulation of β-catenin, but also differ in several aspects, including transcriptional regulation and activity[12,15]. This study was motivated by the intriguing difference of axin and conductin in terms of their subcellular distribution. Whereas axin is predominantly present in cytoplasmic puncta, mediated by DIX–DIX domain interaction[16,17], conductin is diffusely distributed[11]. This is of functional relevance as DIX-mediated polymerization was shown to be important for the activity of axin in degrading β-catenin[16]. Our study has solved the molecular basis underlying this differential distribution and activity by identifying an aggregation mediating activity (aggregon) present in the RGS domain of conductin, but not axin, which prevents efficient DIX-mediated polymerization. Mutation of this site induced puncta formation of conductin and increased its activity. Interestingly, also axin polymerization is sensitive to RGS-mediated aggregation: transfer of the conductin RGS domain into axin, but not of the aggregon mutated conductin RGS domain, prevented axin polymerization. In line with this, axin

mutations occurring in colorectal and liver cancer have been shown to induce axin RGS aggregation and thereby block polymerization[19]. How the aggregation of the RGS domain prevents DIX interactions is presently unclear. It is conceivable that the DIX domains become spatially separated in the conductin aggregates so that they cannot form ordered head-to-tail polymers essential for β-catenin destruction activity. Alternatively, the highly dynamic polymerization process involving rapid association and disassociation of DIX domains might be impaired by RGS aggregation-mediated reduction of mobility. In this study, polymerization into puncta was observed using ectopically expressed proteins. This puncta formation is considered a meaningful model of endogenous destruction complex assembly, since it was confirmed for endogenous axin[23] and impaired polymerization correlates with reduced inhibition of β-catenin[16,19,23,24].

In vivo, lethality of *Axin1* knockout mice was rescued by *Axin2* knock-in suggesting functional equivalence of the two paralogs[25]. In contrast, several studies described differences between axin and axin2/conductin in vitro[15,26–29]. Here, we add to this list that axin and conductin also differ in their subcellular distribution and their capacity to degrade β-catenin due to an aggregon exclusively present in conductin. These differences might not have been revealed in the in vivo study (i) due to compensatory mechanisms, (ii) because they become important only under specific physiological/pathological conditions, or (iii) because only *Axin2* was knocked in into the *Axin1* locus but not vice versa[25].

Importantly, our data suggest that the aggregon in the conductin RGS domain represents a target to regulate conductin activity by enhancing its polymerization. As a proof of principle we show that expression of free RGS domains or of a small peptide ($P^{182-195}$) centered on the aggregon induced conductin puncta formation, i.e. polymerization. $P^{182-195}$ showed direct interaction with the conductin RGS domain in vitro and reduced RGS-mediated aggregation strongly suggesting that it induces conductin polymerization by saturation of the aggregon. It is conceivable that physiologic RGS interaction partners, like e.g. Gα subunits[30,31], might also regulate conductin activity via this mechanism. Regulation of a signaling pathway through the polymerization status of involved scaffold proteins, as described here for Wnt signaling and conductin, offers an elegant mechanism to control signaling activity, and could also be deployed in other pathways.

Activation of human conductin, i.e. axin2, by increasing its polymerization holds potential for cancer therapy. Elevated β-catenin signaling is a hallmark of colorectal cancer with more than 90% of cancers exhibiting mutations of Wnt signaling components which activate the pathway[3]. Moreover, inhibition of

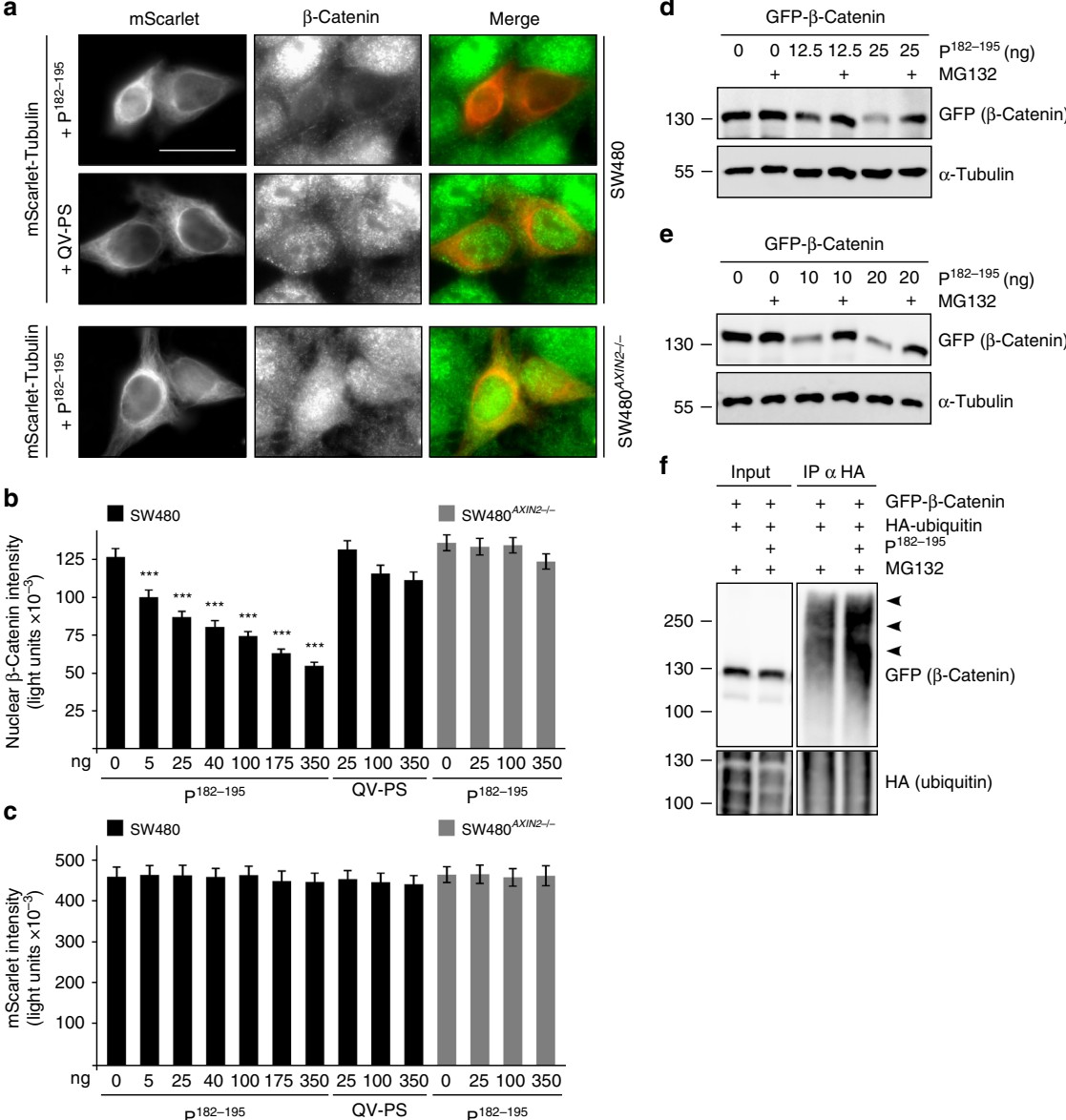

**Fig. 6** P[182–195] induces proteasomal degradation of β-catenin. **a** Immunofluorescence staining of endogenous β-catenin (green) in SW480 and SW480 *AXIN2* knockout cells co-transfected with P[182–195] or its QV-PS mutated analog together with mScarlet-tubulin (red) to visualize transfected cells. Scale bar: 20 μm. **b, c** Quantification of nuclear β-catenin (**b**) or mScarlet (**c**) fluorescence intensities in individual cells of four independent experiments as in **a**. Results are mean ± SEM ($n = 80$); ***$p < 0.001$ (Student's *t*-test). Data distribution is shown in Supplementary Fig. 10. **d, e** Western blotting for GFP and α-tubulin (loading control) in lysates of DLD1 (**d**) or HEK293T cells (**e**) transfected with indicated constructs, which were untreated or treated with the proteasome inhibitor MG132 (DLD1: 10 μM for 6 h; HEK293T: 2.5 μM for 2 h). Given nanograms (ng) of the peptide refer to the transfection of a 12 well. **f** Western blotting for GFP and HA in lysates (Input) of HEK293T cells transfected with indicated constructs, and after precipitation of HA-ubiquitin conjugated proteins from these lysates (IP α HA). Arrowheads point to polyubiquitinated GFP-β-catenin. HA blots show similar overall ubiquitination (Input) and similar precipitation of ubiquitinated proteins (IP) in both samples. Source data are provided as a Source Data file

Wnt/β-catenin signaling was shown to interfere with colorectal cancer growth[22,32]. As a β-catenin target gene, *AXIN2* is highly expressed in human colorectal cancer[12] but does not suffice to block cancer growth. In the majority of colorectal cancer cases this is due to reduced destruction complex activity after mutations of APC. Remarkably, however, expression of axin proteins above a certain threshold, e.g., by overexpression or stabilization by tankyrase inhibition suffices to degrade β-catenin in the absence of APC and blocks colorectal cancer proliferation[22]. It is tempting to speculate that functional activation of axin2 by interfering with aggregate formation might likewise pass the threshold for β-catenin degradation. Indeed, expression of the

peptide P[182–195] alone induced degradation of β-catenin and blocked growth of colorectal cancer cells. P[182–195] most likely acts through activation of endogenous axin2 because peptide activity was reduced after *AXIN2* knockdown and knockout, or mutation of the aggregation sequence in the *AXIN2* gene. Therefore, pharmaceutical stimulation of axin2 polymerization might offer a treatment strategy for colorectal cancer and other Wnt dependent types of cancer. Moreover, this approach would allow exploiting the increased levels of axin2 in tumors as compared to normal tissue to increase the specificity for cancer cells. We have shown that the cell permeable synthetic version of the peptide (P[182–195]-R9) reduces Wnt/β-catenin signaling and inhibits cell growth

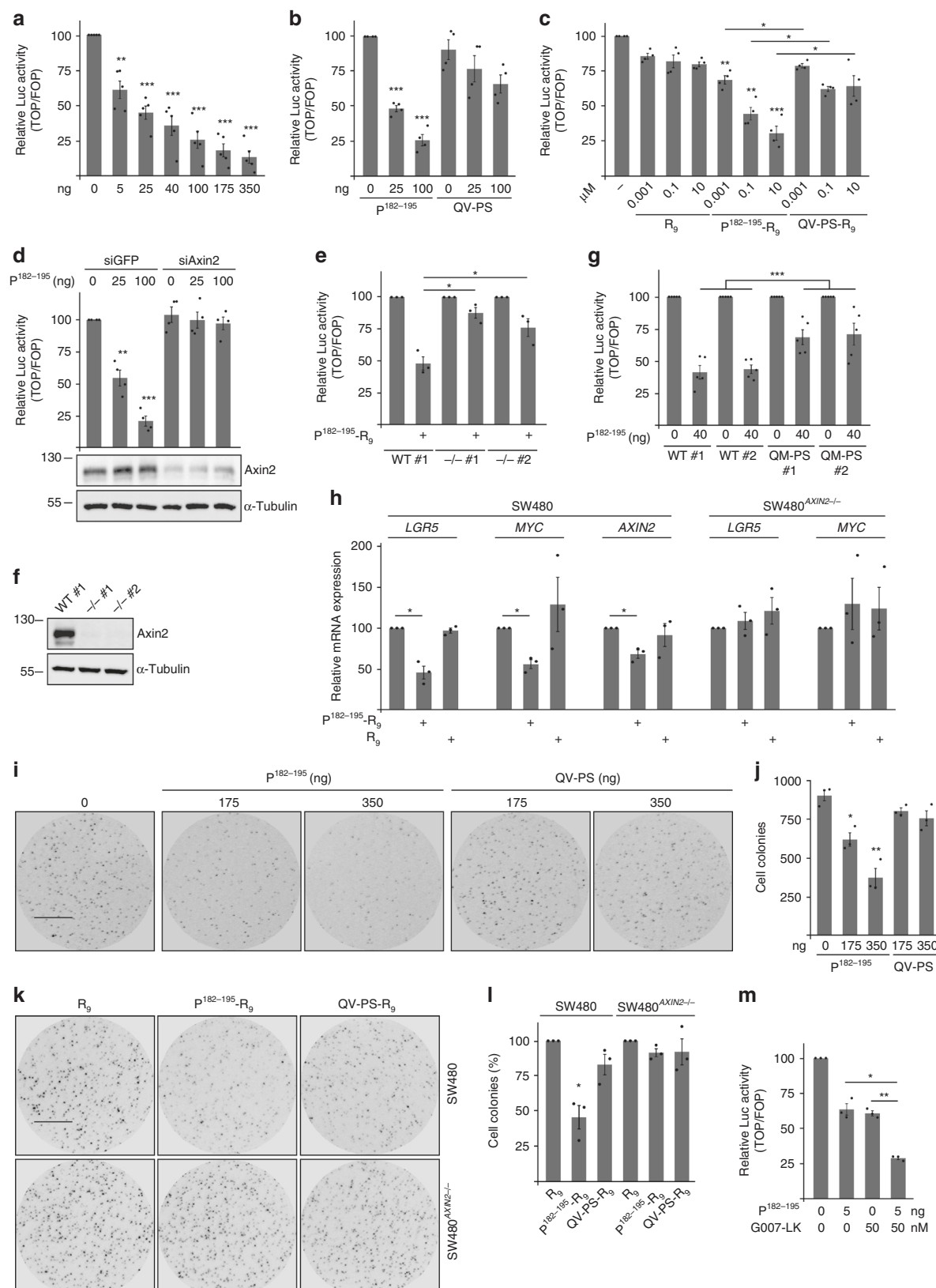

when added to colorectal cancer cells. In addition, expression of the peptide synergized with tankyrase inhibitors that stabilize axin2[22], suggesting molecular options for combined pharmaceutical interference with Wnt signaling.

## Methods

**Cell culture, transfection, small chemicals, and peptides.** DLD1, HEK293T, SW480, and U2OS cells were grown in Dulbecco's Modified Eagle Medium (DMEM) supplemented with 10% fetal calf serum and 1% penicillin/streptomycin in a 10% $CO_2$ atmosphere at 37 °C, and subcultured according to recommendations

**Fig. 7** P[182–195] inhibits Wnt signaling and blocks growth of colorectal cancer cells. **a–e, g, m** Luciferase activity (TOP/FOP) in SW480 cells transfected with indicated amounts of P[182–195] or the QV-PS mutated control (**a, b**) in SW480 cells which were untreated or treated with indicated concentrations of the synthetic peptides R$_9$ (control), P[182–195]-R$_9$, or its QV-PS mutant (QV-PS-R$_9$) for 48 h (**c**), in SW480 cells transfected with P[182–195] together with siRNA against GFP (control) or against axin2 (**d**), in parental SW480 cells (WT #1) and two *AXIN2* knockout clones (−/− #1, 2) which were untreated or treated with 0.1 µM P[182–195]-R9 for 48 h (**e**), in parental SW480 cells (WT #1), a WT axin2 control clone (WT #2), and two axin2 QM-PS mutated clones (QM-PS #1, 2) transfected with P[182–195] (**g**), in SW480 cells transfected with P[182–195] and/or treated with 50 nM of the tankyrase inhibitor G007-LK overnight (**m**). Western blot below **d** shows efficient axin2 knockdown. Results are mean ± SEM (*n* = 5 [**a, g**], *n* = 4 [**b, c, d**], *n* = 3 [**e, m**]). **f** Western blot shows absence of axin2 in the *AXIN2* knockout clones used in **e**. **h** Relative mRNA expression of the β-catenin target genes *LGR5*, *MYC*, and *AXIN2* normalized to *GAPDH* in SW480 cells or SW480 *AXIN2* knockout cells which were untreated or treated for 48 h with 10 µM of indicated peptides. Results are mean ± SEM (*n* = 3). **i, k** Cell colonies grown for 96 h from SW480 cells which were transfected with increasing amounts of P[182–195] or its QV-PS mutated analog together with GFP, sorted by GFP expression and sparsely plated (**i**), or from SW480 cells or SW480 *AXIN2* knockout cells which were treated with 10 µM of indicated synthetic peptides (**k**). Cells were stained by ethidium bromide incorporation and visualized with UV light. Scale bar: 0.5 cm. **j, l** Automated quantification of colony numbers from three independent experiments as in **i** (**j**) or **k** (**l**). Results are mean ± SEM (*n* = 3). *p < 0.05, **p < 0.01, ***p < 0.001 (Student's *t*-test). Given nanograms (ng) of the peptide refer to the transfection of a 12 well. Source data are provided as a Source Data file

by ATCC where the cell lines originated from. HEK293T and U2OS cells were transfected with polyethylenimine, and DLD1 and SW480 cells with Lipofectamine2000 (Invitrogen), according to the manufacturer's instructions. Wnt3a conditioned medium was prepared as originally described[33]. G007-LK was obtained from AdooQ Bioscience and MG132 from Sigma-Aldrich. The 9x arginine control peptide (R$_9$), P[182–195]-R$_9$ and P[182–195]-R$_9$ QV-PS were obtained from Thermo Scientific.

**CRISPR/Cas9-edited cell lines**. SW480 *AXIN2* knockout cells and *AXIN2* Q188P, M189S cells were generated by using the CRISPR/Cas9 system, as described elsewhere[34]. In short, a top scoring guide RNA according to the CRISPR Design tool provided by the Zhang Lab (crispr.mit.edu) targeting exon 2 of human *AXIN2* (guide RNA sequence: CGAGATCCAGTCGGTGATGG; score: 84) was cloned in the PX458 Cas9/GFP/guideRNA-expression vector purchased from Addgene (#48138). The generated PX458 plasmid was transfected in SW480 cells together with a single-strand template for homology directed repair (HDR). The HDR repair template contained two homology arms, and was designed to introduce the desired QM-PS mutation and a silent SpeI restriction site for screening. Twenty-four hours post transfection, GFP-positive cells were sorted and seeded as one cell per well in 96-well plates. From the outgrowing colonies, DNA was isolated, amplified by PCR (for: GGTTTGCCTGCAATGGATTCAGG; rev: ACTGTCAAC AGTTTCCGTGGACC), and test digested with SpeI. SpeI digestion-positive clones were expanded and the homozygote mutation of QM to PS was verified by sequencing. When generating *AXIN2* knockout cells, outgrowing cell colonies were initially screened for loss of axin2 expression by western blot, and *AXIN2* knockout clones were verified by sequencing.

**Plasmids and siRNAs**. Expression plasmids for mScarlet-tubulin[35], APC[36], Flag-Axin, Flag-Cdt, GFP, GFP-β-catenin, GFP-Axin (human), GFP-Axin (rat), GFP-AxinM3, GFP-Axin2, GFP-Cdt, GFP-Axin$^{CdtDIX}$, and GFP-Cdt$^{AxinDIX}$, and the siRNAs against GFP and human conductin/axin2 are published[15,37]. The other plasmids were generated by classical molecular biology methods (GFP-AxinΔ1-209, GFP-CdtΔRGS, GFP-Cdt 2–345, GFP-RGS, GFP-SAMP [APC aa 1516–1591 including the first SAMP repeat; Q61315], HA-Cdt, HA-CdtM3, HA-Cdt 2–345, and peptide P[182–195]). The exchanges to clone expression plasmids for chimeric axin conductin (AC) proteins (GFP-Axin$^{CdtRGS}$, GFP-Cdt$^{AxinRGS}$, GFP-AC370–840, GFP-AC1–318, and GFP-AC197–318) are based on Clustal Omega alignments[38], and the remaining conductin amino acids are indicated for all chimeric proteins in Supplementary Fig. 3. Point mutations were introduced by site-directed mutagenesis (GFP-CdtM3$^{AxinRGS}$, GFPCdtΔRGSM3, GFP-Cdt Y93S, GFP-Cdt F112R, GFP-Cdt Q188P V189S, GFP-Cdt Q188P V189S M3, GFP-Cdt Q188P, GFP-Cdt V189S, GFP-Cdt L99R, GFP-Cdt L99R Q188P V189S, GFP-Axin$^{CdtRGS}$ QV-PS, GFP-Axin2 Q188P M189S, GFP-Cdt 2–345 QV-PS, GFP-Cdt 2–345 Y93S, GFP-Cdt 2–345 F112R, and peptide P[182–195] QV-PS). All generated plasmids were verified by sequencing.

**Immunofluorescence**. Cells were fixed either in −20 °C cold methanol for 5 min (Figs. 1b, d, 2c, f, 4a, 5b, e, 6a and Supplementary Figs. 1, 3 and 4b) or for 10 min at RT in 3% PFA (Supplementary Fig. 1), permeabilized with 0.5% Triton X-100, and blocked with medium to prevent unspecific binding of antibodies. Then, cells were incubated consecutively with primary antibodies (m α APC [1:100], ab58 Abcam/m α β-catenin [1:200], sc-7963 Santa Cruz Biotechnologies/rb α Flag [1:300], F7425; rb α HA [1:200], H6908 Sigma-Aldrich) and fluorochrome-conjugated secondary antibodies (goat α mouse-Cy2/Cy3, goat α rabbit-Cy2/Cy3 [Cy2: 1:200/Cy3: 1:300], Jackson ImmunoResearch) for 1 h each. Stained cells were analyzed with an Axioplan II microscope system (Carl Zeiss) using a Plan-NEOFLUAR ×100 1.30 NA oil objective. Images were acquired at RT with a SPOT RT Monochrome camera (Diagnostic Instruments), and, when required, fluorescence

intensities were quantified from images acquired at constant exposure times using MetaMorph analysis software (Carl Zeiss).

**Western blot, pulldown, and immunoprecipitation**. Cells were lysed 24 h or, in case of siRNA experiments, 48 h after transfection in Triton X-100-based lysis buffer (150 mM NaCl, 20 mM Tris-HCl pH 7.5, 5 mM EDTA, 1% Triton X-100, Roche protease inhibitor cocktail, or the luciferase assay buffer stated below) in hypotonic lysis buffer (20 mM Tris-HCl, pH 7.5, 1 mM EDTA, Roche protease inhibitor cocktail) when analyzing β-catenin levels (Fig. 6d, e), or in 2× sample buffer (150 mM Tris-HCl pH 6.8, 6 mM EDTA, 12.5% glycerol, 3% SDS, 3% β-Mercaptoethanol, 0.2 mM bromophenol blue), when required due to marked differences in solubility (Fig. 4d). For pulldown (Fig. 2g) or immunoprecipitation (Fig. 6f) assays, the lysate was aliquoted in three samples which were supplemented with equal amounts of indicated GST-tagged proteins and glutathione beads (pulldown), or the lysates were supplemented with indicated antibodies and G/A beads (Santa Cruz Biotechnologies) (immunoprecipitation). After rotation of the samples at 4 °C, bead-associated proteins were precipitated, washed, and eluted from beads by boiling in 2× sample buffer. Cell lysates (and eluates) were analyzed by PAGE under denaturing (SDS-PAGE, denaturing of samples by heat, SDS and β-mercaptoethanol, and SDS in buffers for gel pouring and gel run) or, for Fig. 3a, b, upper panel and Fig. 5m, non-denaturing, native conditions (no sample denaturation and buffers without SDS). During denaturing SDS-PAGE, proteins are separated only according to their sizes, whereas under native conditions also protein folding and/or protein–protein interaction can affect the electrophoretic mobility. After gel electrophoresis, proteins were transferred onto a nitrocellulose membrane (VWR) which was probed with indicated primary (m α APC [1:1000], 11814460001 Roche/m α β-actin [1:1000], A5441; rb α Flag [1:1000], F7425; rb α HA [1:1000], H6908 Sigma-Aldrich/rb α Axin2 [1:1000], 2151S; m α GST [1:1000], 2624S CellSignaling/rat α α-tubulin [1:1000], MCA77G Serotec) and respective horseradish peroxidase (HRP)-conjugated secondary antibodies (goat α mouse/rabbit/rat-HRP [1:2000], Jackson ImmunoResearch). Protein bands were visualized with a LAS-3000 (FUJIFILM) by emission of light upon HRP-catalyzed oxidation of luminol. Uncropped scans of blots are provided in the Source Data file.

**Dot blot**. For dot blotting, nitrocellulose membrane pieces (VWR) were placed individually in wells of a 96-well plate, spotted with H$_2$O or 8 nmol of the indicated synthetic peptides, air-dried, and blocked with 5% skimmed milk in phosphate-buffered saline (PBS) to reduce unspecific protein binding. Afterwards, membrane pieces were incubated individually with indicated cell lysates or in vitro-translated proteins (see below), and membrane-bound GFP proteins were visualized using primary and HRP-conjugated secondary antibodies, as described above for the western blot. GFP intensities were quantified with AIDA 2D densitometry.

**Sucrose density gradient ultracentrifugation**. For sucrose density gradient preparation, 2 ml of a 50% (w/v) sucrose solution was overlaid with 2 ml of a 12.5% (w/v) sucrose solution in a 13 × 51 mm centrifuge tube (Beckman Coulter), and placed horizontally for 3 h at RT to allow equilibration of a linear density gradient[39] before overlaying with the sample. Centrifugation was performed using a MLS-50 rotor in an Optima MAX Ultracentrifuge (Beckman Coulter) at 50,000 r.p.m. for 16 h at 25 °C. Afterwards, 20 fractions of equal volumes were collected from top to bottom, and indicated fractions were analyzed by western blotting (see above).

**Recombinant proteins**. Sequences encoding amino acids 74–219 of rat axin, the respective amino acids 67–208 of mouse conductin, and the point mutated versions (QV-PS, L106R) were cloned into the pGEX-6P-1 expression vector. Expression of GST-tagged proteins was induced by IPTG in transformed Rosetta (DE3) bacteria within their logarithmic growth phase at 37 °C for 3 h. Then, bacteria were pelleted and lysed in STE buffer (10 mM Tris-HCl pH8, 1 mM EDTA, 150 mM NaCl)

supplemented with lysozyme (100 µg/ml), protease inhibitors, and either 1% SDS or 1% Triton X-100, as indicated in Fig. 3d. From the Triton X-100 lysates, GST-tagged proteins were precipitated with glutathione beads, washed, and eluted from the beads by adding free glutathione. Lysates (SDS and Triton X-100) and eluates were analyzed by SDS-PAGE (also see western blot) and consecutive staining of the gel with Coomassie Brilliant Blue R-250 (Fluka).

For in vitro preparation of GFP-RGS, the coding sequence was cloned in a T7 expression vector via standard molecular biology methods. Transcription and translation was performed using PURExpress (NEB), a cell-free transcription/translation system reconstituted from the purified components necessary for *Escherichia coli* translation, following the manufacturer's guidelines.

**Luciferase reporter assay**. DLD1, HEK293T, or SW480 cells were transfected with a luciferase expression plasmid either under the control of a β-catenin-dependent promoter (TOP-flash, TCF optimal) or with a β-catenin-independent promoter (FOP-flash, TCF far-from-optimal), together with a β-galactosidase expression plasmid. Co-transfection of other plasmids or siRNA, and treatment with Wnt3a, G007-LK, or synthetic peptides for 24–48 h is indicated in the respective figure legends and figures. Cells were lysed (25 mM Tris-HCl pH 8, 2 mM EDTA, 5% glycerol, 1% Triton X-100, 20 mM DTT), and the luciferase and β-galactosidase activities were measured as emission of light upon luciferin dec-arboxylation in a Centro LB 960 Microplate Luminometer (Berthold technologies) and as release of yellow ortho-Nitrophenol upon ortho-nitrophenyl-β-galactoside hydrolysis with a Spectra MAX 190 (Molecular Devices), respectively. TOP and FOP luciferase activities were normalized to the respective β-galactosidase activity to correct for small variations in transfection efficiency between samples, before calculating TOP/FOP ratios. TOP/FOP assays were performed in technical duplicates.

**MTT cell growth assay**. DLD1 and SW480 cells were transfected with GFP-Axin, GFP-Conductin, or GFP-Conductin QV-PS and 24 h post transfection low GFP-positive cells were sorted (gating strategy see Supplementary Fig. 8) and seeded as 2500 cells per well in 96-well plates. After growth periods of 0, 72, 96, and 120 h, MTT (Sigma-Aldrich) was added at a final concentration of 0.5 mg/ml and cells were incubated for 4 h at 37 °C to allow MTT cleavage in the living cells, before dissolving the produced MTT formazan by adding 100 µl isopropanol with 0.04 N HCl per well. The formazan absorbance was measured with a Spectra MAX 190 (Molecular Devices) at 570 nm and the background measured at a reference wavelength of 690 nm was subtracted. The background-free formazan absorbance is directly proportional to the number of living cells. The assay was performed in technical sextuplicates (Fig. 4e) or quadruplicates (Supplementary Fig. 9).

**Colony formation assay**. In case of transient peptide expression, SW480 cells were transfected with GFP either together with a control plasmid, a plasmid encoding the peptide P[182–195] or the QV-PS mutated peptide, and 24 h post transfection low GFP-positive cells were sorted and seeded as 3000 cells per well in six-well plates. In case of synthetic peptide treatment, cells were pre-treated with 10 µM of the indicated peptide for 24 h before reseeding as 3000 cells per well in six-well plates in DMEM supplemented with 1% fetal calf serum and 10 µM of the indicated peptide. After 96 h, cell colonies were fixed with 3% PFA, stained with ethidium bromide (50 µg/ml in PBS), and visualized in a UV gel documentation system (Herolab), where images were acquired at constant settings. Using the MetaMorph analysis software (Carl Zeiss), numbers of colonies were quantified from these images in an automated fashion by applying a threshold for light objects. The assay was performed in technical duplicates.

**qRT-PCR analysis of Wnt target genes**. Following cell treatment with indicated concentrations of synthetic peptides for 48 h, whole cellular RNA was isolated (RNeasy mini kit, Qiagen), cDNA was synthesized (AffinityScript QPCR cDNA Synthesis Kit; Agilent Technologies) and quantitative PCR was performed in a CFX96 Real-Time System (Bio-Rad) in technical triplicates. mRNA expression of the Wnt/β-catenin target genes *AXIN2*, *LGR5*, and *MYC* was normalized to the housekeeping gene *GAPDH*[10,12,40]. *AXIN2* expression was not analyzed in *AXIN2* knockout cells since the premature stop codon could affect mRNA stability.

*AXIN2*: for: CCTCAGAGCGATGGATTTCGGG; rev: CCAGTTCCTCTCAGCAATCGGC

*GAPDH*: for: GTCAAGGCTGAGAACGGGAAGC; rev: GGACTCCACGACGTACTCAGCG

*LGR5*: for: CTTCCAACCTCAGCGTCTTCACC; rev: GTCAGAGCGTTTCCCGCAAGAC

*MYC*: for: CTTCTCTGAAAGGCTCTCCTTGC; rev: CGAGGTCATAGTTCCTGTTGGTG

**Statistical information**. To probe data sets for statistical significant differences two-tailed Student's *t*-tests were performed. Depending on the experimental setup, tests were performed in a non-paired (Figs. 4b, e, 6b, Supplementary Fig. 4d) or paired fashion (all other experiments). All *n* values used to calculate the statistics are provided in the respective figure legends. Based on the nature of the assays and graphical data assessment, normal data distribution was assumed, which was not

formally tested due to the low information content of those tests for most of the given sample sizes. When required, statistical significance is indicated by asterisks (*$p < 0.5$, **$p < 0.01$, ***$p < 0.001$), as indicated in figure legends.

**Reporting Summary**. Further information on research design is available in the Nature Research Reporting Summary linked to this article.

## Data Availability

The source data underlying Figs. 1e, 2d, g, 3a–d, 4b–e, 5c, d, f–m, 6b–f and 7a–h, j, l, m and Supplementary Figs. 4c-e, 7, 9, 10 and 11a-c, e are provided as a Source Data file. All the other data supporting the findings of this study are available within the article and its Supplementary Information, and from the corresponding author upon reasonable request. A reporting summary for this article is available as a Supplementary Information file.

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

## Acknowledgements
The authors thank Yves Muller for providing the pGEX-6P-1 vector, the core-unit Cell-sorting with Immunomonitoring (Friedrich-Alexander University Erlangen-Nürnberg), Laura Gardill and Andreas Pelz for initial experimental contributions, and Gabriele Daum for excellent technical assistance. This study was funded by a grant from the Friedrich-Alexander University Erlangen-Nürnberg Interdisciplinary Center for Clinical Research to D.B.B. (J58).

## Author Contribution
D.B.B. was involved in conception and design, acquisition and interpretation of data, and writing the manuscript. M.B. was involved in acquisition of data. M.V.H. was involved in interpretation of data. J.B. was involved in conception and design, interpretation of data, and writing the manuscript.

## Additional information

**Competing interests:** The authors declare no competing interests.

