## [Peer Review File · Nature Communications]

Reviewers' comments:

Reviewer #1 (Remarks to the Author):

The work described by Bernkopf et al focuses on the difference in polymerizing activity of Axin and its paralogue conductin (Axin2). Both proteins mediate important tumor suppressor activity within the Wnt signaling pathway and their mutational inactivation is linked to developmental defects as well as cancer formation. In current models, the assembly of Axin in cytosolic punctate structures is mediated by the self-polymerizing activity of the C-terminal DIX domain, which facilitates the assembly of the destruction complex and promotes the degradation of beta-catenin. Strikingly, conductin is much less capable of forming polymerized structures in cells, limiting its activity as a pathway suppressor. In the present study, the authors present convincing evidence that the biophysical properties of the RGS domain are an important determinant of Axin polymerizing behavior. They uncover that a short aggregation-prone motif, present in the conductin RGS domain, interferes with DIX domain-mediated polymerizing activity. Importantly, based on the conductin aggregation-prone motif the authors derive a peptide that antagonizes the inhibitory activity of the conductin RGS domain, allowing the protein to form polymerized structures and downregulate Wnt pathway activity in colorectal cancer cells.

The manuscript is written in a comprehensible manner, the data are of high quality and the figures are clear and well-presented. The scientific results present novel insights in the structure-function relationships of Axin and conductin and provide interesting leads for improved conductin suppressor function in cancer therapy. The main conclusions are strong enough to warrant publication in Nature Communications.

The following points need to be addressed:

- It would be helpful to include a 3D structural model of the conductin RGS domain and map the mutated residues (Fig. 2). Which of these mutations are expected to destabilize the RGS domain? This should be discussed.
- Fig. 2C: Did the authors check the importance of single mutations as well (Q188, V189)?
- Fig 2F: APC colocalization with the QV-PS conductin variant could be mediated via alternative interaction sites (Pronobis, eLife 2015), or indirectly via bcatenin in punctate structures. Can the authors confirm this finding via co-immunoprecipitation or (preferably) via a direct method (ITC, FA) using the isolated conductin RGS domain?
- Fig 3A on its own is not very helpful, it would be more informative to model the PS residues onto the whole RGS domain.

- Fig. 3C: Can the authors exclude interactions with other proteins as an explanation for the observed higher molecular weight in native gels?
- Fig. 4D and E: expression blots should be shown
- Fig. 4D: Can the authors comment on the observed TOP/FOP activation in the absence of Wnt?
- Fig. 5D: Can the authors check expression levels of the peptide?
- Page 8-9: Since the authors have generated purified peptide – can they proof a direct interaction with conductin RGS in vitro?

Minor points:

- The authors call their functional peptide ‘the peptide’ at multiple places. It would be better and more recognizable to come up with a name for it (e.g. residue numbers of the RGS domain)
- Page 1: substitute “titel” by “title”
- Quantification of Δ RGS in Fig. 1E is missing
- Explain in the legend of Fig. 2B the meaning of :/.
- Remove in page 7 line 1 the word “of”
- Fig. 4E: Can the authors compare proliferation levels of Axin, is this similar to QV-PS?
- Order of figures: 5F>5A
- Fig. 5C could just be mentioned in the text
- Page 21 in legend Fig. S1: Remove conductin
- Page 21 in legend Fig. S2: Maybe authors can explain the construct schemes already in S2 and not refer to a posterior figure
- Fig. S3B: Is there also the vice versa domain swap of AC?
- Martino-Echarri et al. (2016) showed that tankyrase inhibition leads to endogenous Axin puncta. Can they see puncta formation with SW480 cells AXIN2 Q188P M189S when treating with tankyrase inhibitor G007-LK?
- State antibody dilutions used for WB and IF

Reviewer #2 (Remarks to the Author):

This manuscript by Bernkopf et al describes how polymerisation in Axin is regulated via the RGS domain. Starting from the observation that Axin but not Conductin forms puncta in cells they were able to define an aggregation site within the RGS domain that is responsible for this difference in behaviour. They went on to define a peptide from that RGS site that is capable of changing the distribution of Conductin in the cell from diffuse to punctate. This peptide was capable of growth reduction in colorectal cancer cells as well as inhibiting β -catenin dependent transcription.

The manuscript is well written and the findings are explained in a concise manner. The discovery of an aggregation site in Conductin that inhibits polymerisation of Conductin via the DIX domain is novel and of interest to the Wnt signalling field. The possibility of targeting this aggregation in Axin2, which is usually upregulated in colon cancer seems a promising target for colon cancer and should be further explored. This possibly for novel cancer therapy gives it a broader appeal beyond the Wnt signalling field.

Overall the work is very convincing, however there are some further experiments that will strengthen the papers claims and scope (see below). The materials and methods are very detailed and precise, which should enable reproducibility of experiments.

Recommended improvements:

- To enhance the generality of the finding an analysis of Axin and Conductin RGS region throughout evolution would be helpful. The authors do mention that the aggregation is conserved between rat and human, but more distant species would enhance the importance of this site (e.g. fish, birds, reptiles and more distant species). An alignment figure would be great to have in the supplement, possibly with a few selected species analysed by TANGO too.

Is the aggregation always present in Conductin, but never in Axin?

- Figure 3A and B should be improved.

The pdb used for the model needs to be specified (1dk8 or 1emu I assume). The space filling model is not ideal for making the point of surface exposure, a simple cartoon model should suffice. This then also makes Fig. B a little obsolete (Figure 3B could be moved to the supplement), especially as the pdb file is of high quality and shows the proline to be surface exposed. This brings me to one of my main questions. Did the authors try the Q>P mutation only? Did this show the same effect as the double mutation? My speculation is that the proline causes a kink in the helix (see attached file,

proline in magenta), which would cause a more general change within the RGS fold. This could create a changed binding site. It is also noteworthy that a previous mutation L106R (Anvarian et al) lays in close proximity to the PS site (attached file Leu in cyan). Upon initial inspection the mutation L106R would change the hydrophobic environment that abutts the helix comprising the PS residue. So both Conductin QV and L106R might be causing the same effect.

- The TANGO aggregation score for Conductin (Fig. 2A) and human Axin2 (Fig. S4A) is markedly different. Conductin has a score of about 15, whereas Axin2 is ten-fold lower at about 1.5. The authors should reflect this in the text by changing following sentence on page 6: "Aggregation site III is also conserved in human axin2 where a QM sequence replaces the QV present in conductin (Fig. S4 A)."

Is this change of aggregation score also true for other more distant species (related to first point)?

- For Figure 5 A and B I would recommend to compare the levels of Flag-Cdt and GFP-RGS wt and mut proteins by Western blot and place this information below the bar graph of Fig 5B.

- The amounts of peptide used are different in the various experiments – can the authors elaborate on this please. Why should TOPFlash experiments not require the same amount of peptide than the experiments of b-catenin reduction? If this is due to experimental setup (e.g. 6-well vs. 12-well) then labeling the peptide by molarity might help explain this.

If this is not the case then I would feel more comfortable with the data if all concentration ranges are shown for all experiments in Fig 6 (0, 5, 25, 40, 100, 175 and 350ng). Are there any toxicity issues at high concentrations?

- As the experiment in Figure 6H is central to a claim made in this paper of this peptide being a possible colorectal cancer therapeutic I feel that this experiment needs a few more controls. As a further negative control the mutated peptide-R9 and/or a scrambled peptide-R9 should be included. If possible, a dose dependency for the cell penetrating peptide-R9 should be shown too.

- In the discussion about how the aggregation of the RGS prevents DIX interactions one point could be the reduction of mobility of the polymer. DIX domains associate and disassociate in a rapid manner, if RGS aggregates prohibit this, then puncta formation might be inhibited. The idea of spatial separation is possible, but the region between the RGS and the DIX domain is flexible and over 500 residues long. This should give enough flexibility and distance to allow DIX-DIX interaction. FRAP experiments might allow the authors to address this in the future.

Minor changes:

GSK3 should be glycogen synthASE kinase 3

“In line WITH...” or “In line WITH THIS...” is the correct use

Brand names should be in caps: e.g. Addgene, Lipofectamine, Thermo Scientific...

Throughout the paper I would recommend to use “distribution” rather than “localization” to describe the diffuse or punctate nature of Conductin and Axin.

Label Fig5 A with Flag-Cdt and GFP-2-345

Add 14-mer to following sentence: “Vector-based transient expression of this 14-mer peptide efficiently induced polymerization of co-expressed conductin in about 80% of the cells”

Figure 4A legend: Dashed, not dotted lines

Open questions:

How is the aggregon of Conductin regulated endogenously?

Is the Axin mutant L106R described in Anvarian et al showing the same effect as QV-PS in Conductin?

Reviewer #3 (Remarks to the Author):

In this study, Bernkopf and colleagues discovered an aggregon in the RGS domain of conductin, but not Axin, which inhibits DIX domain-mediated polymerization. Mutation of this aggregon enhances polymerization and b-catenin degradation activity of conductin. Authors also identified a short peptide that promotes polymerization of conductin through masking the aggregon. The peptide increased b-catenin degradation and inhibited proliferation of colorectal cancer cells. Overall, this is a nice study. Identification of an aggregon in the RGS domain of conductin is very interesting, and it nicely explains the intriguing difference of axin and conductin in terms of their subcellular

localization. Further, the study also suggests a novel strategy of promoting b-catenin degradation in colorectal cancer cells. I have following suggestions for authors to improve the manuscript.

1. Authors used migration on the native gel to demonstrate aggregation of the RGS domain of conductin. Since this is a critical point of the manuscript, I think it can be further strengthened using alternative methods such as gel filtration.
2. Authors demonstrated that the peptide increased puncta formation of conductin (Fig. 5) without showing that the peptide actually inhibits RGS domain mediated aggregation as proposed. This should be directly tested as done in Fig. 3C.
3. Fig. 6 should be improved. It remains possible that the peptide decreased b-catenin expression and inhibited cells proliferation through off-target activities. Does the peptide increase ubiquitination of b-catenin? Ideally, authors should generate Axin2 knockout DLD1 and SW480 cells, and examine the effect of the peptide on b-catenin expression/ubiquitination, expression of b-catenin target genes, and cell proliferation. Adding additional colorectal cancer cell line such as DLD1 in this section would further strengthen the conclusion.
4. Why would the peptide decrease the luciferase activity in QM-PS cells (Fig. 6E)?
5. What is the transfection efficiency of the peptide-expressing plasmid in Fig. 6F? It is somewhat surprising that growth inhibitory effect can be seen using transient transfection. It implies the transfection efficiency of this experiment is much higher than what was shown in Fig. 6A. Why would not authors use cell penetrating peptide (WT vs QV-PS) for this experiment?
6. In an earlier study of the group, authors showed that the protein expression of conductin is significantly lower than that of axin in SW480 and DLD1 cells (PMID: 25380820, Fig. 1B). It seems to be surprising that reactivating conductin has such a strong effect on b-catenin degradation in SW480 cells, considering that the expression of axin is higher than that of conductin.
7. Costantini group has shown that conductin can functionally replace axin using knockin mice (PMID: 15899843), suggesting that axin and conductin are functionally equivalent. Authors should mention and discuss this paper.
8. Authors should acknowledge that polymerization of axin and conductin was shown using ectopically expressed proteins, and this has not been directly demonstrated for the endogenous proteins.

Reviewer #1 (Remarks to the Author):

The following points need to be addressed:

- It would be helpful to include a 3D structural model of the conductin RGS domain and map the mutated residues (Fig. 2). Which of these mutations are expected to destabilize the RGS domain? This should be discussed.

We generated a 3D structural model of the conductin RGS domain based on the crystal structure of the axin RGS domain (1DK8) using SWISS-MODEL, and mapped the relevant mutated residues of aggregation site III for which we have seen functional consequences, as requested (Fig. 2 E). As suggested by reviewer #2 the model is shown in cartoon style. Conclusions drawn from this 3D model were supported by calculations of relative surface accessibility. This information was moved to the supplement, also following suggestions by reviewer #2 (Fig. S6).

Whether these mutations are expected to destabilize the RGS domain is now addressed in the result section of the manuscript as follows:

“A 3D structural model of the conductin RGS domain based on the crystal structure of the highly similar axin RGS domain (Fig. 2 E), and calculations of relative surface accessibility (Fig. S6) showed that the key residues of aggregation site III (Q188 and V189) are exposed at the surface of the RGS domain rather than being buried within the structure. Therefore, mutation of these residues is not expected to generally destabilize the RGS domain. Furthermore, these residues were mutated into the amino acids which are present at the respective positions in axin indicating their compatibility with correct RGS domain folding (Fig. 2 B).”

In line with this conclusion drawn from the RGS domain structure, mutation of the critical residues (QV to PS) did not abolish binding of APC to the RGS domain (see point to Fig. 2 F below) again strongly suggesting that the mutations do not generally destabilize the RGS domain.

- Fig. 2C: Did the authors check the importance of single mutations as well (Q188, V189)?

We now generated and analyzed the single mutants Q188P and V189S showing that both individual mutations suffice to induce polymerization of conductin, yet, with a markedly reduced efficiency compared to the QV-PS double mutation (Fig. 2, C and D) indicating that both residues are important for function of aggregation site III.

- Fig 2F: APC colocalization with the QV-PS conductin variant could be mediated via alternative interaction sites (Pronobis, eLife 2015), or indirectly via β-catenin in punctate structures. Can the authors confirm this finding via co-immunoprecipitation or (preferably) via a direct method (ITC, FA) using the isolated conductin RGS domain?

As requested, we confirmed interaction of the QV-PS mutated RGS domain with APC by showing that the QV-PS mutation does not impair binding of the isolated conductin RGS domain to the isolated RGS binding site of APC (SAMP repeat) in pulldown assays (Fig. 2 G) indicating that the RGS folding is not impaired. In this experiment, binding via alternative interaction sites or indirectly via β-catenin can be excluded.

- Fig 3A on its own is not very helpful, it would be more informative to model the PS residues onto the whole RGS domain.

As suggested in point one for Fig. 2, we now generated a 3D structural model of the conductin RGS domain showing that Q188 and V189 are solvent exposed residues at the RGS domain surface, which can potentially mediate protein-protein interaction. By including this model in Fig. 2 E, Fig. 3 A became obsolete and was removed.

- Fig. 3C: Can the authors exclude interactions with other proteins as an explanation for the observed higher molecular weight in native gels?

To address this point, we purified sufficient amounts for Western blotting of recombinant GST-RGS Cdt and GST-RGS Cdt QV-PS from bacteria, although GST-RGS Cdt was hardly soluble which already suggests aggregation in the absence of cellular proteins (Fig. 3 D). Subsequent native gel electrophoresis showed that GST-RGS Cdt is less mobile than GST-RGS Cdt QV-PS (Fig. 3 B) indicating that no other cellular proteins are required for the formation of the observed higher molecular weight complexes.

- Fig. 4D and E: expression blots should be shown

Fig. 4 D: Expression blot was added.

Fig. 4 E: When sorting the transfected cells for the MTT assay, we made sure to analyze cells expressing GFP-conductin and GFP-QV-PS at comparable levels by using identical sorting gates. Instead of showing an expression blot, we now included the respective sorting gates (Fig. S8 B). This analysis is even more meaningful in this context, since even a similar variation of expression between the cells within the gates can be seen, in contrast to assessing only the overall expression level by Western blot.

- Fig. 4D: Can the authors comment on the observed TOP/FOP activation in the absence of Wnt?

HEK293T cells do show a basal Wnt/ β -catenin signaling activity possibly by autocrine Wnt stimulation, which is also suggested by the expression of the highly specific target gene conductin/axin2 (data not shown). However, this activity is rather low: absolute TOP/FOP ratios of unstimulated HEK293T cells were about 4, whereas those of colorectal cancer cells like SW480 cells and DLD1 cells were about 70 and 40, respectively.

- Fig. 5D: Can the authors check expression levels of the peptide?

We did not come up with an applicable method to determine the expression levels of the non-tagged peptide. Unfortunately, tagging with flag reduced the activity of the peptide by about 50% and, in addition, did not generate a convincing staining signal for quantification in IF experiments (data not shown).

We assume that this question was meant to strengthen that there is a qualitative difference between the peptide and the QV-PS mutant by showing equal expression. Since the peptide and its mutant are transcribed from identical vectors and only differ in two amino acids, we are positive that they are expressed equally. In addition, new experiments performed with synthetic peptides where the amounts can be controlled show qualitative differences between the functional peptide (P¹⁸²⁻¹⁹⁵-R₉) and the newly synthesized QV-PS mutant (QV-PS-R₉) (e.g. Figs. 5, G-L, 7 C, S11 A).

- Page 8-9: Since the authors have generated purified peptide – can they proof a direct interaction with conductin RGS in vitro?

We can now proof direct interaction of the purified peptide with the RGS domain in vitro via Dot blot assays. RGS domain-containing proteins in cell extracts or obtained by in vitro translation (direct binding) specifically interacted with the purified peptide but not the QV-PS mutated peptide (Fig. 5, G-L).

Minor points:

- The authors call their functional peptide ‘the peptide’ at multiple places. It would be better and more recognizable to come up with a name for it (e.g. residue numbers of the RGS domain)

Thanks for the suggestion. We call the peptide now P¹⁸²⁻¹⁹⁵.

- Page 1: substitute “titel” by “title”

done

- Quantification of Δ RGS in Fig. 1E is missing

Requested quantification was added.

- Explain in the legend of Fig. 2B the meaning of :/.

done

- Remove in page 7 line 1 the word “of”

done

- Fig. 4E: Can the authors compare proliferation levels of Axin, is this similar to QV-PS?

Proliferation of SW480 cells expressing conductin QV-PS was similar or even a bit lower compared to axin expressing cells (Fig. S9). Cell numbers were compared 96 h after seeding, a time point at which we reproducibly observed differences between conductin and conductin QV-PS expressing cells.

- Order of figures: 5F>5A

done

- Fig. 5C could just be mentioned in the text

done

- Page 21 in legend Fig. S1: Remove conductin

We are sorry but we don't know why “conductin” should be removed from the figure legend, since it is part of the figure.

- Page 21 in legend Fig. S2: Maybe authors can explain the construct schemes already in S2 and not refer to a posterior figure

done

- Fig. S3B: Is there also the vice versa domain swap of AC?

Yes, there are vice versa domain swaps of the key protein parts/domains shown in Fig. S3 B: e.g. **a** (AC319-840) and **b** (AC1-318), or **e** (AC69-196) and **f** (AC1-68/197-840) represent pairs of vice versa domain swaps. For consistency, all chimeric constructs are named “AC” (in alphabetical order of axin and conductin) and throughout the manuscript always the amino acids of conductin present in these constructs are indicated.

- Martino-Echarri et al. (2016) showed that tankyrase inhibition leads to endogenous Axin puncta. Can they see puncta formation with SW480 cells AXIN2 Q188P M189S when treating with tankyrase inhibitor G007-LK?

Yes, we do see puncta formation of axin2 QM-PS after G007-LK treatment. However, this was also the case for WT axin2, which is line with previously published results (Thorvaldsen et al., 2015). There was a tendency towards increased puncta formation of the QM-PS mutant but the differences are rather subtle, and we prefer not to include these data. Since the major contribution to puncta formation in this experimental setup seems to come from the inactivated tankyrase, which can also polymerize (Martino-Echarri et al., 2016), we are not sure whether one can expect marked differences between axin2 and the QM-PS mutant.

Thorvaldsen, T.E., N.M. Pedersen, E.M. Wenzel, S.W. Schultz, A. Brech, K. Liestol, J. Waaler, S. Krauss, and H. Stenmark. 2015. Structure, Dynamics, and Functionality of Tankyrase Inhibitor-Induced Degradasomes. *Mol Cancer Res.* 13:1487-1501.

- State antibody dilutions used for WB and IF

done

Reviewer #2 (Remarks to the Author):

Recommended improvements:

1.

- To enhance the generality of the finding an analysis of Axin and Conductin RGS region throughout evolution would be helpful. The authors do mention that the aggregon is conserved between rat and human, but more distant species would enhance the importance of this site (e.g. fish, birds, reptiles and more distant species). An alignment figure would be great to have in the supplement, possibly with a few selected species analysed by TANGO too.

Is the aggregon always present in Conductin, but never in Axin?

We performed the suggested sequence alignment paralleled with TANGO analysis (Fig. S5). Indeed, the aggregon is always present in conductin but never in axin. Even in frog conductin/axin2, which shows the lowest sequence conservation, a peak of the TANGO score at aggregation site III suggests functional conservation of the aggregon.

The following was added to the result section of the manuscript:

“Extension of this analysis to evolutionary more distant species representing the five vertebrate classes, i.e. mammals, birds, reptiles, amphibians and fish, revealed rather low sequence conservation of the conductin QV motif (Fig. S5 A). However, the conductin/axin2 TANGO scores of all analyzed species peak at the position of aggregation site III (Fig. S5 B). In axin, the central proline residue is strictly conserved among all species accompanied by an aggregation propensity of 0 at this position (Fig. S5, A and B). These data suggest that aggregation site III of conductin/axin2 is functionally conserved at least among vertebrates, and that axin aggregation is prevented, possibly by the conserved proline residue.”

2.

- Figure 3A and B should be improved.

The pdb used for the model needs to be specified (1dk8 or 1emu I assume). The space filling model is not ideal for making the point of surface exposure, a simple cartoon model should suffice. This then also makes Fig. B a little obsolete (Figure 3B could be moved to the supplement), especially as the pdb file is of high quality and shows the proline to be surface exposed.

As suggested, we indicated the pdb used for the model (1DK8) in the figure legend, changed the model style from space filling to cartoon with highlighting the critical residues as sticks, and moved the relative surface accessibility calculation to the supplement (Fig. S6). Due to suggestions by reviewer #1 we are now showing a model of the whole conductin RGS domain calculated by SWISS-MODEL using the axin structure 1DK8 as template. This model was moved to Fig. 2 (Fig. 2 E).

This brings me to one of my main questions. Did the authors try the Q>P mutation only? Did this show the same effect as the double mutation? My speculation is that the proline causes a kink in the helix (see attached file, proline in magenta), which would cause a more general change within the RGS fold. This could create a changed binding site.

We now generated and analyzed the individual point mutants. Individual Q188P mutation was sufficient to induce polymerization (Fig. 2 C f). Thus, it is well possible that a proline caused kink in the helix splits a straight aggregation site in WT conductin in two parts thereby creating a changed binding site. Beyond this alteration, we do not believe in more general, secondary changes within the RGS fold, since (i) the kinked helix is present in WT axin and does not impair RGS folding and (ii) the

QV-PS mutated conductin RGS domain still interacts with APC indicating functional RGS domain folding (Fig. 2 F and G).

In addition to this Q/P position, also other residues contribute to the functionality of the conductin aggregon because the single V189S mutation was also sufficient to induce polymerization (Fig. 2 C g), and the efficiency of the individual point mutations (Q188P and V189S) was markedly lower compared to the QV-PS double mutant (Fig. 2 D). The V189S mutation might result in the loss of hydrophobic interaction surfaces required for aggregation. By the way, V189M mutation, as present in human axin2, did not alter conductin distribution (data not shown).

It is also noteworthy that a previous mutation L106R (Anvarian et al) lays in close proximity to the PS site (attached file Leu in cyan). Upon initial inspection the mutation L106R would change the hydrophobic environment that abutts the helix comprising the PS residue. So both Conductin QV and L106R might be causing the same effect.

Judging from the close proximity of L106 to the hydrophobic PS helix environment, it is indeed intriguing to speculate that the L106R mutation affects the PS helix in a way to function similar as the QV helix in conductin. However, Anvarian et al. provide convincing evidence that the axin L106R cancer mutation activates the otherwise inaccessible aggregation site II (in our nomenclature) which is distinct from the QV-PS site. We can now show that mutation of L99 to R, which corresponds to the L106R mutation in axin, abolishes polymerization of the conductin QV-PS mutant (Fig. 2 C i). This can be either explained by L99R affecting the now existing PS or, following Anvarian et al.'s line of thinking, by the activation of aggregation site II by L99R which compensates for the loss of aggregation site III upon QV to PS mutation. We now included these data because they support the idea of regulating polymerization of axin proteins via aggregation of the RGS domain. Since we are not investigating the detailed consequences of the L106R axin cancer mutation here, it is out of the scope of this study to analyze whether the L106R mutation in addition influences the PS site as indicated by the close proximity noticed by the reviewer, and we agree with the reviewer (see "open questions") that this remains an interesting point to be addressed in the future

3.

- The TANGO aggregation score for Conductin (Fig. 2A) and human Axin2 (Fig. S4A) is markedly different. Conductin has a score of about 15, whereas Axin2 is ten-fold lower at about 1.5. The authors should reflect this in the text by changing following sentence on page 6: "Aggregation site III is also conserved in human axin2 where a QM sequence replaces the QV present in conductin (Fig. S4 A)."

We changed the sentence as follows:

"In line with conservation of aggregation site III, the TANGO score of human axin2 where a QM sequence replaces the QV present in conductin shows a peak at this position (Fig. S4 A). Although the TANGO score of axin2 is markedly lower compared to conductin, the change from a diffuse distribution of human axin2 to puncta formation upon QM to PS mutation demonstrated functional conservation of aggregation site III (Fig. S4, B and C)."

Is this change of aggregation score also true for other more distant species (related to first point)?

TANGO scores of newly analyzed species were all between the observed scores from human (~ 1.5) and mouse (~ 15), with e.g. the score of frog being somewhere in between (~ 3) (Fig. S5 B). Thus, our mutational analysis for human axin2 and mouse conductin shows functionality of the aggregation site for both extrema of the TANGO score. The TANGO score of axin was markedly lower for all species (see first point),

and the axin2 score dropped to 0 upon replacing the central amino acids (QM/QV/TS) by PS in silico (Fig. S5 B).

4.

- For Figure 5 A and B I would recommend to compare the levels of Flag-Cdt and GFP-RGS wt and mut proteins by Western blot and place this information below the bar graph of Fig 5B.

The recommended Western blot was added, showing that polymerization-inducing GFP-2-345 is rather less expressed than the inactive GFP-2-345 QV-PS mutant (Fig. 5 D, compare also Fig. 3 A). Expression of Flag-Cdt was equal.

5.

- The amounts of peptide used are different in the various experiments – can the authors elaborate on this please. Why should TOPFlash experiments not require the same amount of peptide than the experiments of b-catenin reduction? If this is due to experimental setup (e.g. 6-well vs. 12-well) then labeling the peptide by molarity might help explain this.

If this is not the case then I would feel more comfortable with the data if all concentration ranges are shown for all experiments in Fig 6 (0, 5, 25, 40, 100, 175 and 350ng). Are there any toxicity issues at high concentrations?

The peptide amounts in IF experiments were indeed higher than in TOPFlash experiments for the technical reason to ensure sufficient co-expression of the peptide with GFP which is extremely well expressed (in our hands). To bypass this issue, we now used mScarlet-tubulin to label transfected cells which is less well expressed than GFP. We also increased the sensitivity of the assay by specifically measuring nuclear β -catenin, which is the transcriptionally active fraction. As suggested, we performed this new IF β -catenin degradation experiment (Fig. 6, A-C) in parallel with the TOPFlash experiment (Fig. 7 A) for all peptide concentrations (0, 5, 25, 40, 100, 175 and 350 ng) and observed comparable dosage-dependency, which in our opinion solves this critical issue raised by the reviewer. (Following a suggestion by reviewer #1 the peptide is now called P¹⁸²⁻¹⁹⁵.)

Since degrees of repression in the TOPFlash measurements were highly similar between previous and new experiments at a given amount of transfected peptide (compare Fig. 6 C, D, E and F of the old manuscript version to new Fig. 7 A), we abstained from repeating every single TOPFlash experiment with all plasmids amounts which would blow up figures without giving additional information.

When analyzing the peptide effect on cell proliferation in colony formation assays, we prefer to stay with the higher amounts because this assay takes several days and the peptide plasmid is diluted out with every cell division.

We did not observe unspecific toxicity issues at high concentrations, for instance the FOP or β -galactosidase values were not reduced when expressing 175 or 350 ng of the peptide.

6.

- As the experiment in Figure 6H is central to a claim made in this paper of this peptide being a possible colorectal cancer therapeutic I feel that this experiment needs a few more controls. As a further negative control the mutated peptide-R9 and/or a scrambled peptide-R9 should be included. If possible, a dose dependency for the cell penetrating peptide-R9 should be shown too.

We agree with the high importance of the synthetic peptide and strongly elaborated this part. As suggested, we included the QV-PS mutated peptide-R₉ as further negative control. Of note, QV-PS-R₉ was significantly less active than WT peptide P¹⁸²⁻¹⁹⁵-R₉.

(Fig. 7 C and Fig. S11 A). Furthermore, we observed dose dependency also for the cell penetrating peptide-R9 (Fig. 7 C and Fig. S11 A). These experiments were performed in SW480 and DLD1 cells, following the suggestion of reviewer #3 to include a second cell line. Finally, the cell penetrating peptide was almost inactive in *AXIN2* knockout clones (Fig. 7 E).

7.

- In the discussion about how the aggregation of the RGS prevents DIX interactions one point could be the reduction of mobility of the polymer. DIX domains associate and disassociate in a rapid manner, if RGS aggregates prohibit this, then puncta formation might be inhibited. The idea of spatial separation is possible, but the region between the RGS and the DIX domain is flexible and over 500 residues long. This should give enough flexibility and distance to allow DIX-DIX interaction. FRAP experiments might allow the authors to address this in the future.

Thank you for mentioning this alternative explanation. A respective sentence was added to the discussion.

“Alternatively, the highly dynamic polymerization process involving rapid association and disassociation of DIX domains might be impaired by RGS aggregation-mediated reduction of mobility.”

Minor changes:

GSK3 should be glycogen synthASE kinase 3

changed

“In line WITH...” or “In line WITH THIS...” is the correct use

Thank you for the hint. This was changed throughout the manuscript.

Brand names should be in caps: e.g. Addgene, Lipofectamine, Thermo Scientific...

changed

Throughout the paper I would recommend to use “distribution” rather than “localization” to describe the diffuse or punctate nature of Conductin and Axin.

Thank you for this suggestion. We changed it throughout the manuscript.

Label Fig5 A with Flag-Cdt and GFP-2-345

done

Add 14-mer to following sentence: “Vector-based transient expression of this 14-mer peptide efficiently induced polymerization of co-expressed conductin in about 80% of the cells”

done

Figure 4A legend: Dashed, not dotted lines

changed

Open questions:

How is the aggregation of Conductin regulated endogenously?

This is an interesting open question to be addressed in the future. As briefly mentioned in the discussion, we speculate that endogenous binding partners of the RGS domain like e.g. G-proteins might play a role here.

Is the Axin mutant L106R described in Anvarian et al showing the same effect as QV-PS in Conductin?

Please refer to major point two part three for our detailed answer.

Reviewer #3 (Remarks to the Author):

1. Authors used migration on the native gel to demonstrate aggregation of the RGS domain of conductin. Since this is a critical point of the manuscript, I think it can be further strengthened using alternative methods such as gel filtration.

We agree that aggregation of the RGS domain is a critical point of the manuscript, which is worth to be strengthened. As alternative method, we performed ultracentrifugation through a sucrose density gradient. We observed that efficient sedimentation seen with Cdt 2-345 was markedly reduced by the QV-PS mutation (Fig. 3 C) indicating that the formation of higher molecular weight complexes via aggregation of the RGS domain in WT Cdt 2-345 is prevented upon mutational inactivation of aggregation site III.

2. Authors demonstrated that the peptide increased puncta formation of conductin (Fig. 5) without showing that the peptide actually inhibits RGS domain mediated aggregation as proposed. This should be directly tested as done in Fig. 3C.

As requested, we analyzed the influence of the peptide on RGS aggregation using native gel electrophoresis. Co-expression of the peptide reduced the higher molecular weight complexes formed by Cdt 2-345 (Fig. 5 M, arrow) and increased lower molecular weight complexes (Fig. 5 M, arrowheads) suggesting that the peptide actually inhibits RGS domain-mediated aggregation. (Due to a suggestion by reviewer #1 the peptide is now called P¹⁸²⁻¹⁹⁵.)

In line with this, we were also able to show direct binding of the peptide to the RGS domain in vitro (Fig. 5 G-L; see last major point of reviewer #1).

3. Fig. 6 should be improved. It remains possible that the peptide decreased b-catenin expression and inhibited cells proliferation through off-target activities. Does the peptide increase ubiquitination of b-catenin? Ideally, authors should generate Axin2 knockout DLD1 and SW480 cells, and examine the effect of the peptide on b-catenin expression/ubiquitination, expression of b-catenin target genes, and cell proliferation. Adding additional colorectal cancer cell line such as DLD1 in this section would further strengthen the conclusion.

As requested, we generated SW480 *AXIN2* knockout cells and showed the dependency on axin2 for peptide induced (i) decrease of β -catenin levels (Fig. 6 A-C), (ii) inhibition of β -catenin dependent transcription (Fig. 7 E), (iii) reduction of β -catenin target gene expression (Fig. 7 H), and (iv) inhibition of cell growth (Fig. 7 K and L). As further suggested to strengthen the conclusion, we used DLD1 cells as second colorectal cancer cell line to demonstrate that the effects of the peptide are not restricted to SW480 cells: (i) reduction of β -catenin levels (Fig. 6 D), (ii) inhibition of β -catenin dependent transcription (Fig. S11 A) and (iii) its dependency on axin2 (Fig. S11 B), (iv) reduction of β -catenin target gene expression (Fig. S11 C), and (v) inhibition of cell growth (Fig. S11 D and E) were reproducible in DLD1 cells. Finally, we show that the peptide-induced decrease of β -catenin levels can be rescued by inhibition of the proteasome (Fig. 6 D and E), and that the peptide enhances ubiquitination of β -catenin (Fig. 6 F) indicating that the peptide indeed inhibits Wnt/ β -catenin signaling by enhancing axin2-mediated degradation of β -catenin.

4. Why would the peptide decrease the luciferase activity in QM-PS cells (Fig. 6E)?

Notably, the peptide is markedly and significantly less active in these cells, but yes it remains some activity. Our data show that the QM/QV-PS mutation strongly impairs

the activity of the aggregon. However, we don't know whether aggregation is completely prevented by this mutation. One conceivable explanation for the remaining activity of the peptide in QM-PS cells is that the aggregation which is largely but not completely impaired by the QM-PS mutation is further reduced by the peptide e.g. by interfering with adjacent amino acids in the aggregon. Similarly, the QV-PS mutated peptide was strongly and significantly less active but not completely inactive (Fig. 7 B and C).

5. What is the transfection efficiency of the peptide-expressing plasmid in Fig. 6F? It is somewhat surprising that growth inhibitory effect can be seen using transient transfection. It implies the transfection efficiency of this experiment is much higher than what was shown in Fig. 6A. Why would not authors use cell penetrating peptide (WT vs QV-PS) for this experiment?

As correctly judged by the reviewer from the IF images of old Fig. 6 A, the transfection efficiency of SW480 cells is not high enough to directly study cell proliferation in a transiently transfected cell population. Therefore, cells were co-transfected with GFP, as for the IF experiment in Fig. 6 A, and cells of similar GFP intensity were sorted and seeded as 3000 transfected cells per well by the cell sorter (for details see materials and methods section). For clarification, we now included this information in addition in the figure legend.

We now also show inhibition of cell growth by treating the cells with the cell penetrating peptide (Fig. 7 K and L, and Fig. S11 D and E).

6. In an earlier study of the group, authors showed that the protein expression of conductin is significantly lower than that of axin in SW480 and DLD1 cells (PMID: 25380820, Fig. 1B). It seems to be surprising that reactivating conductin has such a strong effect on b-catenin degradation in SW480 cells, considering that the expression of axin is higher than that of conductin.

We appreciate the interest of the reviewer in our previous work. It might indeed seem surprising that activation of conductin polymerization has such a strong impact in the presence of the more abundantly expressed polymerization-competent axin. Either activated conductin and the already present axin together reach a critical threshold, or conductin is of special importance for inhibiting β -catenin signaling in colorectal cancer cells. The latter explanation is supported by an earlier study demonstrating that knockdown of conductin but not of axin prevents inhibition of β -catenin signaling via tankyrase inhibitors, although tankyrase inhibitors strongly stabilize both, axin and conductin (Thorvaldsen et al., 2017).

Thorvaldsen, T.E., N.M. Pedersen, E.M. Wenzel, and H. Stenmark. 2017. Differential Roles of AXIN1 and AXIN2 in Tankyrase Inhibitor-Induced Formation of Degradasomes and beta-Catenin Degradation. *PLoS One*. 12:e0170508.

7. Costantini group has shown that conductin can functionally replace axin using knockin mice (PMID: 15899843), suggesting that axin and conductin are functionally equivalent. Authors should mention and discuss this paper.

We addressed this point by including the following sentences in the discussion: "In vivo, lethality of *Axin1* knockout mice was rescued by *Axin2* knock-in suggesting functional equivalence of the two paralogs (Chia and Costantini, 2005). In contrast, several studies described differences between axin and axin2/conductin in vitro (Bernkopf et al., 2015; Fumoto et al., 2009; Hadjihannas et al., 2012; Hadjihannas et al., 2010; Thorvaldsen et al., 2017). Here, we add to this list that axin and conductin also differ in their subcellular

distribution and their capacity to degrade β -catenin due to an aggregon exclusively present in conductin. These differences might not have been revealed in the in vivo study (i) due to compensatory mechanisms, (ii) because they become important only under specific physiological/pathological conditions or (iii) because only *Axin2* was knocked in into the *Axin1* locus but not vice versa (Chia and Costantini, 2005)."

8. Authors should acknowledge that polymerization of axin and conductin was shown using ectopically expressed proteins, and this has not been directly demonstrated for the endogenous proteins.

Puncta formation, as observed with ectopically expressed proteins, was also shown for endogenous axin (Faux et al., 2008), which could be confirmed in our lab (data not shown). In addition, e.g. reduced activity of the endogenous axin cancer mutant L106R correlated with impaired polymerization of this mutant shown by ectopic expression (Anvarian et al., 2016). Therefore, we believe that polymerization of ectopically expressed axin proteins represents a meaningful model for β -catenin destruction complex assembly, and addressed this issue in the discussion as follows: "In this study, polymerization into puncta was observed using ectopically expressed proteins. This puncta formation is considered a meaningful model of endogenous destruction complex assembly, since it was confirmed for endogenous axin (Faux et al., 2008) and impaired polymerization correlates with reduced inhibition of β -catenin (Anvarian et al., 2016; Faux et al., 2008; Fiedler et al., 2011; Mendoza-Topaz et al., 2011)."

REVIEWERS' COMMENTS:

Reviewer #1 (Remarks to the Author):

The authors have addressed all of my points and have substantially improved the manuscript. The work convincingly presents novel insights into the structure-function relationships of Axin and conductin and provides interesting leads for enhanced conductin suppressor function as a potential therapeutic strategy. I therefore support its publication in current form in Nature Communications.

Reviewer #2 (Remarks to the Author):

The resubmission of Bernkopf et al addresses most of the raised questions from the first submission and further experiments have very much substantiated the original findings. Overall this is a very nice and convincing story and should be published.

There are following issues that might need to be addressed:

- Fig 2G, which is a new panel, needs further controls. Ideally the GST-pulldown would be done against GFP-only or GFP-tagged to an unrelated protein domain, AND GFP-SAMP that has an RGS binding mutation. These controls are important, as GST-RGS Cdt is not as soluble and does not purify as well as the QV-PS mutant, as has been explicitly stated in Fig 3D.

- Fig 7G, can the authors elaborate why cells that ALL have QM-PS CRISPRed into the gene do not show more drastic changes compared to the WT cell lines (approx. 40 vs 60%)?

Minor changes:

- Add labels for the N- and C-terminus of the RGS model in Fig 2E

- page 10 line 3: Do the authors mean Fig 5B and C?

- Was there a reason why cells were cultured at relatively high CO₂ concentrations of 10%?
- mScarlet-tubulin not described in plasmid section
- should be “skimmed” milk used in dot blot
- Fig 5 legend: Scale bars should be described with Immunofluorescent figure, not at end of whole legend
- page 26 line 3: “analog”, what does this mean?
- Fig 6 legend: last sentence should be placed before F)
- use of “left untreated” throughout the manuscript, just use untreated
- some of figures are a somewhat jumbled, for example Fig 7F and could be streamlined for ease of reading

Reviewer #3 (Remarks to the Author):

Authors have addressed my concerns.

Reviewer #1 (Remarks to the Author):

The authors have addressed all of my points and have substantially improved the manuscript. The work convincingly presents novel insights into the structure-function relationships of Axin and conductin and provides interesting leads for enhanced conductin suppressor function as a potential therapeutic strategy. I therefore support its publication in current form in Nature Communications.

Reviewer #2 (Remarks to the Author):

The resubmission of Bernkopf et al addresses most of the raised questions from the first submission and further experiments have very much substantiated the original findings. Overall this is a very nice and convincing story and should be published.

There are following issues that might need to be addressed:

- Fig 2G, which is a new panel, needs further controls. Ideally the GST-pulldown would be done against GFP-only or GFP-tagged to an unrelated protein domain, AND GFP-SAMP that has an RGS binding mutation. These controls are important, as GST-RGS Cdt is not as soluble and does not purify as well as the QV-PS mutant, as has been explicitly stated in Fig 3D.

To address the issue of differential solubility as best as possible, we pre-diluted the better soluble GST-RGS QV-PS to have protein preparations with similar concentrations as starting point for the experiment. The suggested controls merely serve to show that the RGS domain does not bind to GFP, something we never observed in immunoprecipitation experiments, and would thereby further validate the RGS-SAMP interaction, which is a strong, well-characterized and even crystalized interaction (Behrens et al., Science 1998, Spink et al., EMBO J. 2000). Finally, the conclusions drawn from Fig. 2G are supported by Fig. 2F. Therefore, we feel that the suggested controls are not essential here.

- Fig 7G, can the authors elaborate why cells that ALL have QM-PS CRISPRed into the gene do not show more drastic changes compared to the WT cell lines (approx. 40 vs 60%)?

Since the same question was raised by reviewer #3 in the first round of revision, we take the liberty to copy the answer:

Notably, the peptide is markedly and significantly less active in these cells [42 vs 70%], but yes it remains some activity. Our data show that the QM/QV-PS mutation strongly impairs the activity of the aggregon. However, we don't know whether aggregation is completely prevented by this mutation. One conceivable explanation for the remaining activity of the peptide in QM-PS cells is that the aggregation which is largely but not completely impaired by the QM-PS mutation is further reduced by the peptide e.g. by interfering with adjacent amino acids in the aggregon. Similarly, the QV-PS mutated peptide was strongly and significantly less active but not completely inactive (Fig. 7 B and C).

Minor changes:

- Add labels for the N- and C-terminus of the RGS model in Fig 2E

N- and C-terminus have been labelled.

- page 10 line 3: Do the authors mean Fig 5B and C?

Yes, this is what we mean. This point becomes obsolete, as the style of referring to figures had to be changed in general to match the editorial guidelines of *Nature Communications*.

- Was there a reason why cells were cultured at relatively high CO₂ concentrations of 10%?

No, there was no specific reason. These are our regular cell culture conditions.

- mScarlet-tubulin not described in plasmid section

The mScarlet-tubulin plasmid and a respective reference were added to the plasmid section.

- should be "skimmed" milk used in dot blot

Changed

- Fig 5 legend: Scale bars should be described with Immunofluorescent figure, not at end of whole legend

Changed

- page 26 line 3: "analog", what does this mean?

AE for "analogue". In this case we mean the very same peptide except the explicitly stated QV-PS mutation.

- Fig 6 legend: last sentence should be placed before F)

Changed

- use of "left untreated" throughout the manuscript, just use untreated

Changed

- some of figures are a somewhat jumbled, for example Fig 7F and could be streamlined for ease of reading

We agree that the positioning of panel F in figure 7 is not optimal. However, since the panels are unambiguously identifiable through their labels and panel G does not fit at the position of panel F, we prefer to sacrifice the strict order of panels for keeping the whole figure as compact as possible.

Reviewer #3 (Remarks to the Author):

Authors have addressed my concerns.

We thank all three reviewers for their interest in our work, their effort in peer-reviewing our manuscript and especially for their very constructive suggestions for improvement.